# Autochthonous Bacilli and Fructooligosaccharide as Functional Feed Additives Improve Growth, Feed Utilisation, Haemato-Immunological Parameters and Disease Resistance in Rohu, *Labeo rohita* (Hamilton)

**DOI:** 10.3390/ani13162631

**Published:** 2023-08-15

**Authors:** Tanaya Sukul, Zulhisyam Abdul Kari, Guillermo Téllez-Isaías, Koushik Ghosh

**Affiliations:** 1Aquaculture Laboratory, Department of Zoology, The University of Burdwan, Burdwan 713104, West Bengal, India; 2Department of Agricultural Sciences, Faculty of Agro-Based Industry, Universiti Malaysia Kelantan, Jeli Campus, Jeli 17600, Kelantan, Malaysia; 3Advanced Livestock and Aquaculture Research Group, Faculty of Agro-Based Industry, Universiti Malaysia Kelantan, Jeli Campus, Jeli 17600, Kelantan, Malaysia; 4Department of Poultry Science, University of Arkansas, Fayetteville, AR 72701, USA

**Keywords:** functional feed additive, probiotics, prebiotics, FOS, carp, pathogen challenge

## Abstract

**Simple Summary:**

The application of probiotics, prebiotics and synbiotics as functional feed additives has created increasing attention towards environment-friendly aquaculture. In this study, the effects of autochthonous *Bacillus* spp. (probiotics) and fructooligosaccharide (FOS, prebiotic) have been evaluated, either alone or in combination (synbiotics), for an Indian major carp (rohu), *Labeo rohita* fingerlings. The results indicated that the combination of *B. licheniformis* and FOS significantly improved live weight gain, feed utilisation and protease activity in fish compared to the other groups. In general, probiotics- and synbiotics-supplemented groups exhibited improved haematology, serum biochemical profiles and immune parameters compared to the control group. After 90 days of feeding, the fish were experimentally exposed to a pathogenic bacterium, *Aeromonas hydrophila*. Data on the haematology, immunity and stress parameters revealed that synbiotics application boosted immunity and relieved physiological stress in fish. The highest survival among the pathogen-infected fish was recorded with the dietary application of synbiotics consisting of *B. licheniformis* and FOS. In addition, fish fed a combination of *B. methylotrophicus* and FOS in their diets also survived well. Thus, we suggest the application of synbiotics containing autochthonous bacilli and FOS to improve growth, immunity and pathogen resistance in fish.

**Abstract:**

The effects of *Bacillus* spp. (7 Log CFU g^−1^ feed) and fructooligosaccharide (FOS, 1%) as functional feed additives, either alone or in combination, were evaluated in a study on rohu, *Labeo rohita* fingerlings. The fish were fed different diets for 90 days, including a control diet and diets supplemented with FOS, *B. licheniformis*, *B. methylotrophicus* or synbiotic formulations of these. The results showed that the combination of *B. licheniformis* and FOS significantly improved weight gain, feed utilisation and protease activity compared to the other groups. Overall, the groups supplemented with probiotics and synbiotics (*B. licheniformis* + FOS or *B. methylotrophicus* + FOS) showed improvements in haematology, serum biochemistry and immune parameters compared to the control group. After 90 days of experimental feeding, the fish were challenged with pathogenic *Aeromonas hydrophila*, and data on haematology, immunity and stress parameters were collected. The results indicated that the application of *Bacillus* spp. and FOS boosted immunity and resistance to physiological stress in the fish. The highest post-challenge survival rate was observed in fish fed a diet with *B. licheniformis* and FOS, indicating the potential of this particular combination of functional feed additives to enhance growth, immunity and disease resistance in *L. rohita*.

## 1. Introduction

The rising demand to maximise production in intensive and semi-intensive aquaculture requires addressing severe stress in fish, resulting in compromised growth, weakened immune system and increased susceptibility to diseases caused by diverse parasites and pathogens [1,2]. Decreased growth, the emergence of diseases and the subsequent loss in production are adversely affecting both the aquaculture economy and the socio-economic conditions of local fish farmers [3]. Thus, aquafarmers are compelled to apply huge quantities of antibiotics as well as different chemotherapeutic agents to combat against the prevailing stressful situation. However, the indiscriminate use of antibiotics and other traditional drugs has brought about consequences such as altering the symbiotic gut microbial community, increasing the antibiotic resistance and building up the antibiotic residues in fish along with their environment [4,5]. Hence, the application of functional feed additives as natural therapeutic agents has created a growing interest in alternative and environmentally friendly approaches [2,6,7].

As demonstrated in previous reports, probiotics confer benefits that include improving feed utilisation, increasing metabolic efficiency, modulating gut microbiota, synthesising bio-molecules (e.g., amino acids, vitamins, bacteriocins) and creating antagonism against pathogens [8,9,10]. Among these benefits, enhanced nutrient utilisation and immunomodulation are worth mentioning for their benefits in improving the overall health status of the fish. The most commonly used probiotics in aquaculture include lactic acid bacteria and *Bacillus* spp. [11,12]. While applying probiotics as feed supplements, *Bacillus* species are usually taken into consideration owing to their spore-forming ability, which offers greater viability after pelleting and tolerance to gastro-intestinal conditions [11,13]. Although bacilli are ubiquitous in nature, host-associated microbiota are given preference to ensure optimal benefits from the probiotics in analogous environments [2,10,14]. Diverse species of fish gut-associated bacilli, either singly or in combination, have been demonstrated with positive effects on growth performance and disease resistance in a variety of fish species during the last few decades [14].

Unlike probiotics, information on the application of prebiotics in fish is scarce [3,9,11,15], although prebiotics are widely used in humans and other terrestrial animals [11]. Prebiotics are indigestible substances that allow specific alterations in the composition and/or activity of gastrointestinal (GI) microbiota and thus impart a positive effect on the nutrition as well as health status of the host [16]. Several prebiotics, e.g., β-glucan, galacto-oligosaccharide (GOS), mannan-oligosaccharide (MOS) and fructo-oligosaccharide (FOS), have been shown to improve the growth, immune response, survival and protection against pathogens in fish and shellfish species [17,18,19,20]. Among the prebiotics evaluated in fishes, the functional roles of FOS are fairly well established [21]. To date, few studies on the use of FOS in fish are available. For example, previous reports on turbot larvae, *Psetta maxima* [22]; Atlantic salmon, *Salmo salar* [23]; and common carp, *Cyprinus carpio* [24] have suggested that FOS beneficially affects these fish by improving the growth, composition of gut-associated microbiota and immune responses. In addition, synbiotic formulations (probiotics and prebiotics in synergism) are increasingly being evaluated in some fish species and assumed to have beneficial effects of both the agents collectively [3,25,26]. FOS in combination with *B. subtilis* and *B. circulans* was reported to exert positive effects on growth, feed efficiency and non-specific immunity in juveniles of large yellow croaker, *Larimichthys crocea* [11], and rohu, *Labeo rohita* [21], respectively. However, the limited information available concerning the application of synbiotics in aquaculture suggests the need for further investigations on the formulation of novel synbiotic combinations and also investigations to validate their effects on important aquaculture species.

Tropical and subtropical regions are the main production centres of most carps in the world. Among the major carps, *L. rohita* is the most reasonably priced and preferred species cultured in India, comprising about 35% of the total carp production [27]. Bacterial infections are one of the most important causes of disease problems in tropical aquaculture [28]. Among bacterial pathogens, *Aeromonas hydrophila* is the most common and can easily spread through accidental scratches [29,30]. In this context, the present study assessed the likely application of the FOS (prebiotic), either individually or in combination with probiotics (synbiotic), in *L. rohita* fingerlings. In our preceding study, the probiotic potential of the gut-associated *Bacillus licheniformis* (KU556167) and *Bacillus methylotrophicus* (KU556164) isolated from *L*. *rohita* was documented [2,31]. These strains were bile-tolerant, diverse exo-enzymes producers and antagonistic against pathogenic aeromonads [31]. To the best of our knowledge, there is little information on growth as well as immunomodulation induced by the combination of FOS and *Bacillus* spp. other than *B. subtilis*. Therefore, an attempt has been made in the present study to evaluate the effects of the dietary administration of FOS and *Bacillus* spp. (*B. licheniformis* and *B. methylotrophicus*), both individually and in combination as synbiotics, on growth, haemato-immunological parameters and disease resistance capacity against pathogenic *A. hydrophila* in rohu, *Labeo rohita*, fingerlings.

## 2. Materials and Methods

### 2.1. Experimental Fish

Farm-raised healthy fingerlings of rohu, *L. rohita* (weighing ~10 g per fish), without any sign of infection or peripheral abrasion, were obtained from a regional commercial fish farm located at Naihati (22.8929° N, 88.4220° E), West Bengal, India. Specimens were handled and experiments were designed following the approved guidelines of the Institutional Ethics Committees. ARRIVE guidelines were complied with during the experiment. The fish were bathed with 5 ppm potassium permanganate (KMnO_4_) for 5 min and transferred to fibre-reinforced plastic (FRP) tanks (350 L) with steady aeration. The fish were acclimated to the laboratory habitat for 2 weeks. The fish were fed the basal/control diet (Table 1) with ~35% crude protein and fish meal as the main protein source throughout the acclimatisation period (fortnight) and subsequently randomly distributed in the experimental tanks.

### 2.2. Probiotics and Prebiotic Sources

The present study utilised autochthonous *Bacillus licheniformis* (KU556167) and *Bacillus methylotrophicus* (KU556164) as probiotic strains, the probiotic potential of which has been documented in previous reports [2,31]. Pure cultures of the strains were maintained in tryptone soya (TS) broth at −20 °C in 0.85% NaCl with 20% glycerol [32] to provide stable inocula during the experimental period. The broth cultures were grown on TS media plates, and a relationship between colony-forming units (CFU) vs. OD_600_ was established to provide the desired concentration of the probiont in the experimental diets. Natural fructo-oligosaccharide (FOS; Sigma Aldrich, St. Louis, MI, USA) prepared from chicory was used as the dietary prebiotic source. FOS was dissolved (1%, *w*/*v*) in the phosphate buffer and filter-sterilised (0.22 μm) prior to being added to the experimental diets.

### 2.3. Diet Preparation

A basal diet with approximately 35% crude protein was formulated using fish meal, mustard oil cake, rice bran and wheat bran as the major ingredients (Table 1). The basal diet without probiotic, prebiotic (FOS) or synbiotic (probiotics + FOS) incorporation was served as the control diet (C), while the basal diet was mixed with FOS (E1), *B. licheniformis* (E2) and *B. methylotrophicus* (E3), either singly or in combination (*B. licheniformis* + FOS, E4; *B. methylotrophicus* + FOS, E5), to prepare the experimental diets. Probiotic strains were added at 1 × 10^7^ CFU/g diet (7 Log CFU/g). The feeding schedule of the experimental groups is illustrated in Table 2.

The ingredients were ground into a fine powder, sieved through a 400 μm mesh, autoclaved at 15 psi (20 min) and thoroughly mixed with cod liver oil (1%), carboxymethylcellulose (1%, binder) and lukewarm water to prepare the dough. Before water was added, FOS (1%) was added to the prebiotic- and synbiotic-supplemented diets using manual mixing, while bacterial suspensions were unhurriedly added to the probiotic- and synbiotic-supplemented diets in a drum mixer to ensure the homogeneity of the mixture. Prior to pelletisation, a vitamin–mineral premix (Supradyn, Bayer Consumer Care AG, Basel, Switzerland) was added to the diets. Finally, the dough thus prepared was processed through a hand pelletiser to obtain uniform pellets (1.5 mm), oven-dried under aseptic conditions (~60 °C), sieved and stored in a refrigerator (4 °C) until use. Diets with probiotics and synbiotics were prepared every 15-day interval to pledge viability of the bacteria in the diets [5].

### 2.4. Experimental Design

After acclimatisation, 540 rohu fingerlings (mean individual weight 10.17 ± 0.02 g) were distributed in the 18 FRP tanks (6 experimental groups × 3 replicates) of 350 L capacity at a stocking density of 30 fish per tank following a completely randomised design. Fish were fed experimental diets twice daily (09:00 and 14:00 h) at 3% of their total biomass for 90 days. The feeding rate was adjusted every 15th day by batch-weighing. After 4 h of each feeding, the residual feed was siphoned off, oven-dried (100 °C, 24 h) and stored for estimation of the feed conversion ratio [33]. The faecal matter discharged by the fish in each tank was removed daily by pipetting. The experimental fish were reared with seasoned groundwater, steady aeration and a daily exchange of 25% water. The fundamental water quality parameters (dissolved oxygen, 6.5–7.2 ppm, pH 7.4–7.8, salinity 0.35–0.45 ppt, ammonia 0.03–0.06 ppm and temperature 26.4–28.8 °C) were measured every week following the standard methods of the American Public Health Association [34].

After termination of the feeding trial, the fish were subjected to a challenge study against fish pathogenic *Aeromonas hydrophila* (MTCC 1739). *A. hydrophila* was grown in nutrient broth (30 °C, 24 h) and a 7-day lethal dose 50 (LD_50_) was determined [2], which was 10^7^ CFU/fish. After 90 days, 10 fish from each experimental set were injected intraperitoneally (in triplicate) with 100 µL PBS containing 10^7^ CFU of live *A. hydrophila*. The control set was divided into two sub-sets, such that the positive control received live *A. hydrophila*, while the negative control was injected with 100 µL of PBS only. Post-challenge, the fish were kept under strict observation for 10 days and fed the experimental diet. The cumulative mortality was noted, and survivability (%) was calculated. Further, haematological, serum biochemical, immunity and stress parameters were recorded after 10 days of the challenge.

### 2.5. Proximate Composition of Experimental Diets and Fish Carcass

Feed ingredients as well as experimental diets were analysed for proximate compositions (on a dry-weight basis) following the standard methods described by the Association of Official Analytical Chemists [35]. The moisture content was determined by heating the samples primarily at 100 ± 5 °C for 30 min and thereafter at 60 °C until a constant weight was obtained. Crude protein (N × 6.25) was determined using a semi-automatic micro-Kjeldahl digestion system together with distillation and titration units (KjelTRON, Tulin Equipments, Chennai, India). Crude lipid/ether extract (petroleum ether, 50–60 °C) and crude fibre were measured with the Socsplus and Fibraplus systems, respectively (Pelican Equipments, Chennai, India). Ash content was determined by ignition at 550 °C in a muffle furnace. Nitrogen-free extract (NFE) was calculated by deducting the sum of values for crude protein, crude lipid, ash, crude fibre and moisture from 100 [36]. The gross energy of the diets was estimated with a bomb calorimeter (Lab-X, Kolkata, India). Proximate analyses of the whole body of fish (carcass) were performed on a wet weight basis at the beginning and termination of the experiment following the same procedures mentioned above.

### 2.6. Growth Indices

After completion of the 90-day feeding trial, all surviving fish were collected from each tank and weighed to analyse the growth parameters. Growth indices were measured in terms of the specific growth rate (SGR, % day^−1^), weight gain (%), feed conversion ratio (FCR), protein efficiency ratio (PER), apparent net protein utilisation (ANPU%), survivability (%) and condition factor (K) following standard methods delineated by Steffens [37].

### 2.7. Estimation of Digestive Enzymes

Activities of the digestive enzymes in experimental fish were estimated at the beginning and termination of the feeding trial. Gastrointestinal (GI) tracts of the fish from each set were removed under aseptic conditions and washed thoroughly, and a homogenate (10%) was prepared with ice-cold phosphate buffer (0.1 M, pH7). Following centrifugation (10,000× *g*, 4 °C, 10 min), the cell-free supernatant was taken and used as the enzyme extract. Amylase, protease and lipase activities in the GI tract were determined following Bernfeld [38], Walter [39] and Bier [40], respectively. The protein content of the supernatant was estimated following Lowry et al. [41] using bovine serum albumin (BSA) as a standard, and enzyme activities were expressed as units (U).

### 2.8. Collection of Blood Samples

To analyse haematological, serum-biochemical, immune and stress parameters, sampling of blood (as well as mucus, head kidney and hepatic tissue) was performed at the beginning (day 0) and termination of the feeding trial (day 90), as well as after the challenge study. Fish were starved (24 h), and 6 fish from each tank were randomly collected after batch-weighing; hence, a total of 18 fish were sampled per feeding group. Fish were anaesthetised with MS-222 and the tail ablation method was followed to collect blood samples from the caudal vein [42].

Blood samples were taken in heparinised Eppendorf tubes for haematological studies. Serum was separated from the freshly collected blood following Ai et al. [43]. The collected whole blood was allowed to clot at room temperature (~28 °C, 4 h) and centrifuged at 3500 rpm (10 min, 4 °C), and the supernatant was separated as a serum sample and stored at −80 °C for future use.

### 2.9. Collection of Skin Mucus and Head Kidney Leucocytes

Prior to the sampling of blood, skin mucus was collected by keeping the anaesthetised fish in plastic bags containing NH_4_HCO_3_ (100 mM, pH 7.8) following Ross et al. [44]. Mucus samples were centrifuged (12,000× *g*, 15 min, 4 °C), filter-sterilised (0.8–0.22 μm; HiMedia) and stored at −80 °C until further use. Head kidney leucocytes were collected after sampling of blood according to Secombes [45] and modified following Geng et al. [46]. Cell viability (%) was checked using the trypan blue exclusion method, and cell density was adjusted at 1 × 10^7^ cells mL^−1^ using a haemocytometer for successive assay.

### 2.10. Analysis of Haemato-Biochemical Profile

A comprehensive account of the haematological parameters evaluated in this study is presented in Mukherjee et al. [2] and Al-Dohail et al. [47]. The total red blood cells count (RBC) of the heparinised blood was carried out with a haemocytometer and documented as millions of cells per mm^3^ [48]. The total white blood cells count (WBC) was estimated in a Neubauer haemocytometer after dilution with Natt–Herrick solution and recorded as thousands of cells per mm^3^ [48]. Platelets were also counted using a haemocytometer with Reese–Ecker fluid as a dilutant and measured as thousands of cells per mm^3^ [2]. Haemoglobin (Hb) concentration was determined by converting haemoglobin to cyanmethemoglobin using Hb cyanide solution, i.e., Drabkin’s reagent [49]. Haematocrit or packed cell volume (PCV) was presented as percentage packed cell volume (PCV%) measured with micro-haematocrit capillary tubes following the method of Al-Dohail et al. [47]. Erythrocyte Sedimentation Rate (mm/h) was calculated using the Wintrobe Tube protocol [50]. Mean corpuscular volume (MCV; μm^3^), mean corpuscular Hb (MCH; pg/cell) along with mean corpuscular Hb concentration (MCHC; g%) were calculated from erythrocyte count, haematocrit and Hb concentration, respectively, using appropriate formulae [47].

The serum biochemical profile was examined with an automated biochemical analyser (EM Destiny 180, Transasia Bio-Medicals Ltd., Mumbai, India) employing commercial biochemical kits following the manufacturer’s instructions. The parameters studied were total protein, albumin, globulin, cholesterol, triglyceride, LDL (low-density lipoprotein), HDL (high-density lipoprotein), glucose, alkaline phosphatase (ALP), serum glutamic-pyruvic transaminase (SGPT), serum glutamic-oxaloacetic transaminase (SGOT) and lactate dehydrogenase (LDH). Globulin (g dL^−1^) was calculated by deducting albumin (g dL^−1^) from the total protein (g dL^−1^).

### 2.11. Estimation of Immunological Parameters

Serum, skin mucus and head kidney leucocytes were analysed for immunological parameters. Serum and mucus lysozyme activity was detected following Andani et al. [51], using lyophilised *Micrococcus luteus* (ATCC 7468; HiMedia, Mumbai, India) as substrate and egg white lysozyme (Sigma) as standard. The turbidity changes were measured (OD_450nm_) and the lysozyme activity was expressed in units of μg mL^−1^ serum or mucus. The activity of the serum alternative complement pathway (ACP) was analysed following Ai et al. [43] with minor modification [5]. The volume of serum that produced 50% haemolysis (ACH_50_) was estimated with goat red blood cells (OD_414nm_) and the number of ACH_50_ Unit mL^−1^ was expressed. Serum and mucus peroxidase activities were measured using tetramethyl-benzidine hydrochloride (Sigma) with H_2_O_2_ as the substrate [52]. The reaction was blocked with H_2_SO_4_, and OD_550nm_ was recorded as the peroxidase activity. Serum anti-protease activity was measured following Zuo and Woo [53] using azocasein (Sigma) as a substrate that reacted with trypsin solution (Sigma) pre-incubated with serum sample, and OD_450nm_ was recorded. Trypsin inhibition (%) was calculated as [(Control OD—Sample OD)/Control OD × 100] to express the antiprotease activity.

Head kidney leucocytes were analysed for the Phagocytic activity (PA). The fluorescent latex beads were applied to the leucocytes monolayer (2 h, 28 °C), fixed with formalin (10%) and stained with propidium iodide (0.1%) [54]. Phagocytes ingesting latex beads were observed under a fluorescence microscope (DM 1000, Leica Microsystems, Wetzlar, Germany), and PA (%) was calculated [(Phagocytic leucocytes/Total leucocytes) × 100]. Superoxide anion production was assessed by measuring the intracellular O_2_^−^ production through the reduction of nitroblue-tetrazolium (NBT) [55]. Leukocytes maintained in Leibovitz-15 medium (HiMedia) were mixed with NBT (0.2%) and zymosan (Sigma). Intracellular superoxide anion (O_2_^−^) production was indicated by the formation of formazan crystals within the cells and NBT (%) was expressed. The respiratory burst (RB) of the phagocytes was also evaluated by the reduction of NBT that produced intracellular O_2_^−^ radical [43]. The leucocyte suspension was stained with NBT and phorbol 12-myristate13-acetate (PMA), and RB was determined by the intensity of colour development (OD_630nm_).

### 2.12. Estimation of Stress Enzymes

Hepatic (hepatopancreas) tissue was collected, homogenised (10%) in phosphate buffer (50 mM, pH 7.4) and centrifuged (10,000× *g*, 4 °C, 15 min), and the supernatant was used as the crude enzyme source for evaluation of different stress enzymes. The soluble protein content of the crude enzyme extract was determined [41]. Superoxide dismutase (SOD) activity was determined following Nishikimi et al. [56] with minor alteration. The reaction mixture consisting of NBT (300 µM), phenazine methosulfate (186 µM), NADH (780 µM) and crude enzyme extract (100 µL, 120 µg protein) was incubated (30 °C, 15 min); the reaction was terminated with glacial acetic acid; and OD_560nm_ was taken. One unit (U) of SOD activity was defined as the amount of enzyme required to produce 50% inhibition of the NBT reduction, and the enzyme activity was expressed as U/mg protein. Catalase activity was measured following Sinha [57]. The reaction mixture containing H_2_O_2_ (100 μL) substrate and the enzyme extract (100 μL) was incubated (pH 7.4, 30 °C, 15 min). Dichromate acetic acid was added to stop the reaction, and OD_570nm_ was measured. Specific enzyme activity (U) was expressed as μM of H_2_O_2_ utilising mg protein^−1^ min^−1^. Lipid peroxidation was determined using quantification of the malondialdehyde (MDA) level following the method of Ohkawa et al. [58] with minor modifications. Crude extract, SDS (8.1%) and acetic acid (20%) were taken, pH was adjusted to 3.5, and the solution was mixed with thiobarbituric acid (0.8%) and heated at 95 °C (1 h). The absorbance of the final pink-coloured organic layer was recorded at OD_532nm_ against n-butanol blank, and the activity is expressed as nmol MDA/mg protein.

### 2.13. Statistical Analysis

Statistical analyses for all the treatments were performed using the analysis of variance (ANOVA) followed by Tukey’s test [59] using R-Studio 3.6.3 [60]. A comparison was made at the 5% probability levels. The multivariate test, principal component analysis (PCA) and Pearson’s correlation matrix plot were performed by PAST 4.06 for the determination of the correlations among the diverse parameters in different experimental groups. Survival plots (Kaplan–Meier) were analysed using XLSTAT (2007) statistical software and statistical significance was calculated by log-rank test, where *p* < 0.05 was considered significantly different.

## 3. Results

### 3.1. Growth Parameters

The growth performance and feed utilisation of *Labeo rohita* fingerlings fed experimental diets are presented in Table 3. Generally, significantly higher (*p* < 0.05) live weight gain (%) was evidenced in all experimental groups receiving diets with prebiotics (E1), probiotics (E2, E3) or synbiotics (E4, E5) as compared to the control group. The fish receiving diet E4 (*B. licheniformis +* FOS) achieved the highest weight gain (259.72 ± 2.83%) and differed significantly (*p* < 0.05) from the fish fed basal or other experimental diets. The highest SGR (% day^−1^) was also recorded with the fish fed diet E4, although it did not differ significantly from the fish fed diet E5 (*B. methylotrophicus +* FOS). Similarly, the maximum PER along with ANPU were documented in the fish that received diet E4, which was followed by the fish fed diet E5 incorporating *B. methylotrophicus +* FOS. The lowest FCR was detected with the fish fed diet E4, the highest value of which was noticed for the control diet. A significantly higher condition factor (K) was noticed among the treatment groups (E1–E5) compared to the control group. Regarding survivability, there was no significant difference detected among the experimental groups.

### 3.2. Carcass Composition

Proximate composition (% wet weight) of the fish carcass (whole-body) is depicted in Figure 1. Overall, an increasing trend in carcass protein, lipid and ash contents was noticed in the experimental fish that received diets supplemented with probiotics, prebiotics or synbiotics. Compared to the control group, the moisture content significantly (*p* < 0.05) decreased in the experimental fish fed dietary functional feed additives except the group fed diet E1, which contained only FOS as the prebiotic. The highest crude protein in fish carcass was documented in the group fed diet E4 (*B. licheniformis +* FOS), although it was not significantly different (*p* < 0.05) from the group that received diet E5 (*B. methylotrophicus +* FOS). Similarly, a significantly (*p* < 0.05) higher accumulation of carcass lipid and ash was detected with the fish fed diet E4, followed by the groups fed diets E5, E2 and E3. In general, synbiotic- and probiotic-fed groups performed well compared to the control or only prebiotics-fed groups.

### 3.3. Digestive Enzymes

The activities of amylase, protease and lipase in *L*. *rohita* fed experimental diets are depicted in Figure 2. The activities of the studied digestive enzymes improved over the initial (day 0) values in all experimental groups. Overall, significantly higher (*p* < 0.05) activities of the digestive enzymes were recorded in the experimental groups as compared to the control group. Furthermore, enzyme activities were significantly higher in the groups that received synbiotics (E4–E5) compared to the groups fed dietary prebiotics (E1) or probiotics (E2–E3). The maximum amylase activity (11.4 ± 0.09 U) was exhibited by the fish reared on diet E4 (*B. licheniformis +* FOS), although it was not significantly different (*p* < 0.05) from the fish fed diet E5 (*B. methylotrophicus +* FOS). Likewise, the highest protease activity (20.34 ± 0.1 U) was detected in the fish fed diet E4, followed by the group fed diet E5. The fish raised with diet E5 revealed the maximum lipase activity (16.00 ± 0.12 U), followed by the fish fed diet E4.

### 3.4. Haematological Parameters

The haematological parameters of *L. rohita* fed control and experimental diets were investigated on day 0 (beginning) and day 90 (termination of the feeding trial) and after the challenge study, as portrayed in Table 4. Blood cells were noticed to increase over the initial values. At day 90, the fish fed a synbiotic diet E4 (*B. licheniformis +* FOS) were shown to have a significantly (*p* < 0.05) increased blood cells count (RBC, WBC and PCV) and Hb compared to the other groups, including the control, except for the WBC of the fish fed synbiotic diet E5, which was similar to the E4. An increase of 22.77% in the leucocytes was noticed in the fish fed diets E4 and E5 compared to the control group. The maximum platelet count was recorded with the group fed probiotic *B. licheniformis*-incorporated diet (E2), which differed significantly and was followed by the fish fed diet E4 (*B. licheniformis +* FOS) and other experimental groups. Significantly higher MCV was noticed with the fish fed the diet incorporating probiotic *B. methylotrophicus* (E3), which was followed by the group reared on another probiotic, *B. licheniformis* (E2). MCH varied within a narrow range among the experimental groups, demonstrating the maximum values in the groups fed dietary probiotics (E2 and E3). In contrast, the values of the ESR and MCHC had decreased in all experimental groups on day 90 compared to the control group.

The challenge with pathogenic *Aeromonas hydrophila* influenced the haematological parameters in *L. rohita*, although a similar pattern was followed for the major parameters. For instance, the synbiotic-fed group E4 (*B. licheniformis +* FOS) exhibited a significantly (*p* < 0.05) increased blood cell count (RBC, WBC and PCV) and Hb compared to the other experimental groups, including the control. Overall, post-challenge, a decrease in RBC, platelets, Hb, PCV and MCHC and an increase in WBC, ESR and MCV were noticed in all of the experimental groups. Compared to day 90, an increase of 19.45% of the leucocytes was noticed after pathogen exposure in the fish fed synbiotic diet E4.

### 3.5. Serum Biochemical Parameters

The serum biochemical parameters of *L. rohita* fed control and experimental diets were also investigated on day 0, day 90 and after the challenge study and are represented in Table 5. Total protein, along with albumin and globulin levels in the blood, increased significantly (*p* < 0.05) by day 90 in the experimental groups that received dietary probiotics or synbiotics compared to the control and only prebiotic-fed (E1) groups. The maximum serum protein contents and ALP were detected with the fish fed diet E4, which differed significantly from all other groups. While the highest cholesterol, HDL and LDL levels were noticed in the fish reared on diet E5, this did not differ significantly from the fish fed diet E4. Significantly higher serum triglyceride, glucose and SGPT levels were exhibited by the control group; however, the application of dietary probiotics and synbiotics could reduce the values. In contrast, fish fed diets incorporating the probiotic *B. licheniformis*, either solely (E2) or in combination with the prebiotic FOS (E4), were shown to have the significantly lowest SGOT values compared to the other groups. Serum LDH levels varied within a narrow range among the experimental groups.

Post-challenge, increases in the levels of almost all of the serum biochemical parameters were noticed in the experimental fish except albumin and HDL. Compared to the other experimental groups, synbiotic-fed groups (E4 and E5) exhibited significantly (*p* < 0.05) higher levels of serum protein along with globulin after being challenged with *A. hydrophila*. Further, significantly (*p* < 0.05) higher levels of cholesterol, LDL, triglyceride, SGPT, SGOT and LDH were detected post-challenge in the fish fed the control diet (C) or the diet with only prebiotic (E1) compared to the other experimental groups, while significantly improved HDL levels were recorded in the fish that received dietary probiotics (E2–E3) and synbiotics (E4–E5). The maximum ALP was detected in the fish fed diet E4, which differed significantly from all other groups and was similar to the pre-challenge data.

### 3.6. Immune Parameters

Immunological parameters of *L. rohita* fingerlings measured on day 90 and after the challenge study are displayed in Table 6. All of the experimental diets with functional feed additives promoted a significant improvement (*p* < 0.05) in the activities of serum lysozyme, ACP and serum, as well as mucus peroxidase, as compared to the control group at the end of day 90 and also after the *A*. *hydrophila* challenge. Furthermore, significantly improved (*p* < 0.05) activities of mucus lysozyme, serum anti-protease and phagocytes, as well as the RB of the head kidney leucocytes, were recorded before (day 90) and after the challenge study with the fish that received dietary probiotics (E2–E3) and synbiotics (E4–E5). Significantly higher (*p* < 0.05) serum as well as mucus lysozyme activities were noticed in the fish fed a synbiotic diet E4 (21.23 ± 0.15 and 5.05 ± 0.09 μg mL^−1^) compared to the other experimental groups on day 90, representing augmentation by 46.21% and 66.66%, respectively, compared to the control group. A similar trend for the lysozyme activity was shown after the challenge study. The group fed diet E4 resulted in the highest ACP activity, which differed significantly (*p* < 0.05) from other experimental groups, displaying a 52.67% and 61.84% increase in the activity at day 90 and after the challenge study, respectively, compared to the control group.

Similarly, phagocytic activities of the head kidney leucocytes at both time points (on day 90 and post-challenge) were the highest in fish reared on diet E4, which were significantly (*p* < 0.05) higher than the other experimental groups. On day 90, the highest RB activity of the leucocytes was also noticed with the fish reared with E4, which was significantly (*p* < 0.05) different from all other groups and closely followed by group E5. However, post-challenge, RB activity in E4 did not differ significantly from the fish fed diet E5. Serum peroxidase activity on day 90 was significantly higher in the fish fed synbiotic diet E5 compared to the other experimental diets, except for E4, while post-challenge, the maximum serum peroxidase activity was noticed in the fish fed diet E4, which differed significantly from all other groups, including the group fed diet E5. However, mucus peroxidase activities were significantly higher and consistent in both the synbiotic-fed groups (E4 and E5) at both time points compared to the other groups. Among all of the dietary treatment groups, the maximum antiprotease activity was recorded on day 90 in the fish fed synbiotic diet E5, while post-challenge, the fish fed synbiotic diet E4 exhibited the highest activity. Comparison the experimental groups showed that the synbiotic-fed groups (E4 and E5) also showed significantly higher (*p* < 0.05) superoxide anion production by the phagocytes on both day 90 and post-challenge, which was closely followed by the groups that received probiotic-supplemented diets (E2 and E3).

### 3.7. Stress Parameters

Stress parameters, viz., superoxide dismutase (SOD), catalase (CAT) and lipid peroxidation in the liver (hepato-pancreas) tissues of *L. rohita* were recorded on day 0, day 90 and after the challenge study, as presented in Table 7. Overall, SOD and catalase activities were increased and lipid peroxidation was decreased compared to the control group in the fish fed experimental diets supplemented with prebiotics, probiotics or synbiotics (E1–E5). The maximum SOD activities both on day 90 and after the challenge study were recorded in the group fed diet E4 as significantly different from the other groups and exhibited 66.04% and 59.62% enhancement, respectively, compared to the control group. Moreover, the activity of SOD was increased by 29.40% in the fish fed a synbiotic diet E4 following pathogenic infection compared to the value on day 90. Similarly, the highest catalase activities at both time points were recorded in the fish that received diet E4 (*B. licheniformis +* FOS). For lipid peroxidation, the highest concentration of MDA was recorded in the control group at both time points, and it differed significantly from the experimental groups that received dietary functional feed additives, while the concentration of MDA declined with experimental feeding (E1–E5) compared to the control group. The lowest value was detected with the group that received diet E4, which was not significantly (*p* < 0.05) different from diet E5 on day 90. Following *A. hydrophila* exposure, an increase in the MDA concentration was noticed in all the experimental groups compared to day 90. The lowest value was recorded with the fish fed synbiotic diet E4, where a decrease of 34.03% was noticed compared to the control group.

Multivariate tests using principal component analysis were used to correlate weight gain%, FCR, carcass protein, digestive enzymes, stress parameters and selected serum biochemical and immune parameters, as shown in Figure 3a. Two principal components, PC1 and PC2, containing six different experimental groups of correlations between variables, were resolved from the analysis. The first two principal components (PCs) explained 95.76% of the variability present in the data set, with PC1 accounting for 91.33% (on the *Y*-axis) and PC2 for 4.43% (on the *X*-axis) of the total variance. PCA revealed that probiotic- and synbiotic-supplemented groups (E2–E5) exhibited a positive association, with the majority of the parameters studied being the most advantageous in fish that received *B. licheniformis +* FOS as synbiotic supplement (E4). Furthermore, PCA plots were confirmed and quantified by Pearson’s correlation matrix plot, as shown in Figure 3b.

### 3.8. Post-Challenge Survivability

The post-challenge survivability of *L. rohita* fingerlings following exposure to the pathogenic *A. hydrophila* is depicted in Table 8. The results of the challenge study illustrate that dietary synbiotics improved the resistance of *L*. *rohita* against *A. hydrophila* infection. Apart from the negative control group, the highest post-challenge survivability (*p* < 0.05) among the experimental groups exposed to *A. hydrophila* was noticed with the fish fed synbiotic diet E4 (93.33 ± 3.33%), followed by the group fed diet E5 (86.66 ± 3.33%). Thus, survivability (%) in the synbiotic-fed groups (E4 and E5) was increased by 154% and 136%, respectively, compared to the positive control group. Among the groups fed only probiotics (E2 and E3), comparatively better survivability (70 ± 5.77%) was achieved by the fish fed diet E2 with *B. licheniformis*. The lowest survival rate (36.66 ± 8.81%) was observed in the positive control group that did not receive the functional feed additives. The Kaplan–Meier survivorship plots of the fish challenged with *A. hydrophila* were depicted in Figure 4. The *p* value of log-rank test was 0.004, and thus the plots were statistically significant. The dead or moribund fish were noticed with the appearance of typical symptoms of haemorrhagic septicaemia.

## 4. Discussion

The oral administration of functional feed additives has been emphasised in aquafarming to impart positive effects such as growth promotion, disease resistance and improved immunity. Consequently, the supplementation of probiotics, prebiotics and synbiotics as functional feed additives has been documented for diverse fish species in several preceding reports [61,62,63,64,65]. Improvement in digestive enzyme activity, degradation of complex nutrients, synthesis of vitamins, immune modulation, protection against pathogenic organisms and alteration of the microbial community within the gut have been considered the major functions of the probiotic gut bacteria in fish [2]. *Bacillus* spp. belonging to the phyla Firmicutes are predominant within the gut of the IMCs [2,66,67]. Therefore, *Bacillus* spp. have been widely investigated for their probiotic potential in carps [1,2,5]. Oral administration of the prebiotics has also been reported to enhance growth, disease protection and immune function in fish, although the incorporation of the prebiotic feed supplements at different levels has been known to produce varied effects [11,25,68]. Prebiotics such as fructooligosaccharides (FOS) found in various fruits and vegetables are one of the major classes of oligosaccharides that support bacterial growth and thus have been frequently studied in humans as well as other terrestrial animals [11,69]. Although prebiotics cannot be digested by the host itself, they offer positive effects to the host by modulating the growth and activity of one or more of the bacteria within the gastro-intestinal tract. However, the beneficial effects of prebiotics may not last longer than the dietary supplementation period [3]. Therefore, the likely application of the probiotics and prebiotics in combination (as synbiotics) has been suggested to solve this problem. The application of synbiotics in aquaculture has received attention not only due to its beneficial activity towards growth promotion but also for its potential in disease resistance [61]. In consideration of these facts, it is legitimate to provide an investigative report on the dietary application of FOS and *Bacillus* spp., either separately or in combination, to realise the effects of synbiotic-supplemented diets on growth, haematology, non-specific immunity and disease resistance in *L. rohita* challenged with pathogenic *A. hydrophila*.

### 4.1. Growth Performance and Feed Utilisation

In our study, *L. rohita* fingerlings fed prebiotic-, probiotic- and synbiotic-supplemented diets brought about a significant increase (*p* < 0.05) in live weight gain (%), SGR, PER and ANPU compared to the control diet. The maximum growth was recorded with the combined application of *B. licheniformis* and FOS, which was followed by another synbiotic (*B. methylotrophicus* + FOS)-fed group. Our study was in agreement with Zhang et al. [70], who reported improved growth in the golden *pompano*, *Trachinotus ovatus*, fed diets supplemented with *Bacillus subtilis* (5.62 × 10^7^ CFU g^−1^) and FOS (0.2%). Similarly, improvements in growth and feed utilisation were recorded with synbiotic-supplemented diets in large yellow croaker, *Larimichthys crocea* [11]; rainbow trout, *Oncorhynchus mykiss* [71]; European sea bass, *Dientrachus labrux* [72]; Nile tilapia, *Oreochromis niloticus* [73,74]; and rohu, *Labeo rohita* [21]. A preceding report documented the effectiveness of the *B. licheniformis* and *B. methylotrophicus* strains used in the present study as probiotics for rohu [2]. However, the combined application of these strains with FOS further augmented the growth performance of the experimental fish, as recognised in the present report.

In the present study, the proximate composition of the fish carcass exhibited a significant increase (*p* < 0.05) in protein, lipid and ash contents in probiotic- and synbiotic-fed groups. Improved carcass protein and lipid contents, owing to feeding probiotic-supplemented diets, were consistent with the earlier reports on rainbow trout, *Onchorhynchus mykiss* [75]; Caspian kutum, *Rutilus frisii kutum* [76]; and *L. rohita* [2]. For synbiotics, the application of *Entercoccus faecium* (5 × 10^11^ CFU kg^−1^) along with FOS enhanced body protein and lipid contents in rainbow trout [71]. Comparably with the present report, the dietary administration of a synbiotic formulation consisting of *Aspergillus oryzae* and β-glucan improved carcass protein in Nile tilapia [74].

The use of functional feed additives such as pro-, pre- or synbiotics might influence the digestive enzymes that appear to be species-specific and could depend on diverse factors, e.g., gut microbiome composition, environmental condition and life stage, as well as dose, duration and type of administration [77]. In the present study, *L. rohita* fingerlings fed dietary pre-, pro- and synbiotics were recorded with significantly higher (*p* < 0.05) activities of digestive enzymes compared to the control group. Our study was in harmony with that of Ashouri et al. [78], where a remarkable increase in the total protease, trypsin, lipase and α-amylase activities was recorded in the Asian sea bass, *Lates calcarifer* juveniles, due to the administration of sodium alginate or its combination with *Pediococcus acidilactici*. Dietary probiotics or prebiotics have been regarded as potential feed additives owing to their ability to establish a balanced gut microbiota, which could be considered complementary for the supply of appropriate digestive and degradation enzymes, which in turn could enhance the digestibility of the nutrients [14,77,78]. A previous study by Mukherjee et al. [31] indicated the diverse exoenzymes-producing ability of the autochthonous bacilli strains used in the present appraisal. Thus, improved activities of digestive enzymes recorded in the present report could be due to the availability of exoenzymes produced by the gut microbiota in fish that received dietary probiotics and/or prebiotics [5,79]. Although exogenous enzymes produced by the gut bacteria seemed to represent only a small fraction of the total enzyme activity within the gut, the presence of prebiotics and/or probiotics might have stimulated the synthesis of endogenous digestive enzymes, resulting in improved digestive enzyme activities recorded in the present study [5,80]. Further, in the present study, groups fed synbiotics were noticed to have significantly higher digestive amylase, protease and lipase activities than those with the sole applications of FOS or probiotics, exhibiting synergistic effects of the FOS (prebiotic) and probiotic bacilli, as indicated by Ashouri et al. [78]. Analogous observations reporting increased activity of digestive proteases were documented owing to the dietary administration of FOS (2.5 g kg^−1^) and MOS (2.5 g kg^−1^), along with *Bacillus claussi* (10^7^ CFU g^−1^) in Japanese flounder, *Paralichthys olivaceus* [81], and FOS (3 g kg^−1^) and *B. licheniformis* (10^7^ CFU g^−1^) in triangular bream, *M. terminalis* [82].

### 4.2. Haematological Parameters

Haematological parameters are considered as markers to evaluate the physiological status in fish [2,5]. Haematological indices recorded in the present study were consistent with previous reports in carps [2,5,83], with minor variations. Although insignificant with the sole application of FOS, improvements to the haematological indices altogether were noticed in the presently reported study due to the application of probiotics or synbiotics, indicating their potential role in inducing certain physiological responses [84]. The use of probiotics and synbiotics might have stimulated haemopoiesis and thereby improved RBC, WBC and Hb in the experimental fish. Increased RBC and Hb levels might aid in elevating the oxygen-carrying capacity of blood to ensure the normal well-being of the fish [85]. Meanwhile, a decrease in RBCs and Hb levels in pathogen-challenged fish observed in the present study might suggest the depletion/destruction of RBCs owing to leucocytosis activity [55]. The alternation in blood plasma might affect the ESR level in fish, which could be influenced by stress. The decrease in ESR could indicate a reduced risk of infection or inflammation associated with fish, while enhanced ESR noticed in the post-challenged groups may be allied with the fragility of erythrocytes as a result of the aggravation of the induced pathogen [86]. Further, a decrease in MCHC post-infection might also suggest a decrease in RBCs, Hb and PCV due to disturbances that occurred in haematopoietic organs of fish challenged with *A. hydrophila*, which aligned with the previous reports [87,88]. A significant increase in the WBC counts in probiotic- and synbiotic-fed groups, along with a further increase in the WBC levels following *A. hydrophila* challenge, was recorded in the study. This could be indicative of improved innate immunity that might consecutively trigger the primary immune defence mechanism in fish to help them to fight against the pathogens [3]. In agreement with our observation, previous records indicated improved RBC, WBC and Hb levels in Oscar, *Astronotus ocellatus* [89]; Nile tilapia [90,91]; and rohu, *L. rohita* [2,5] fed diets supplemented with either single- or multi-species probiotics. However, reports relating to the effects of dietary prebiotics and synbiotics on the haematology of fish are scarce and have contradictory results. Feeding synbiotics diets in *L. rohita* (1 g kg^−1^; equal quantity of probiotic, *B. subtilis* and GOS) [3] and Caspian brown trout, *Salmo trutta caspius* (2 g kg^−1^ isomaltooligosaccharides, IMOS + 1 g kg^−1^
*Bacillus* spp., BetaPlus^®^) [92], resulted in increased WBCs, which were consistent with the present report. However, on the contrary, the administration of dietary oligofructose [93], and dietary mannan oligosaccharide [94] in the juveniles of beluga, *Huso huso*, could not significantly improve the haematological parameters.

### 4.3. Serum Biochemical Parameters

In contrast to the application of probiotics, the effects of dietary prebiotics or synbiotics on serum biochemical parameters in fish are rarely available [3]. In the presently reported study, the dietary applications of probiotics and synbiotics were associated with a significant increase in serum total protein, globulin and lipid profile (cholesterol, HDL, LDL), along with decreases in glucose, triglyceride and stress enzymes (SGOT, SGPT). Globulin is the chief source of Ig, so enhanced serum globulin might result in immune-stimulatory activity in experimental fish [3]. A further increase in the serum total protein and globulin in the pathogen-challenged fish fed dietary prebiotics or synbiotics might indicate enhanced antibody production, along with improved innate immune function against the pathogen [3,95]. Our study was compatible with the preceding reports in Nile tilapia, *Oreochromis niloticus* [96]; grass carp, *Ctenopharyngodon idella* [97]; and mrigal carp, *Cirrhinus mrigala* [98] fed diets supplemented with probiotics/synbiotic and exposed to the pathogens. In another study, Caspian brown trout fed synbiotics consisting of IMOS and *Bacillus* spp. (BetaPlus^®^) also resulted in increased total protein and globulin [92], which was consistent with the present report. High cholesterol could provide a better health status, while elevated triglyceride levels in the blood might indicate liver damage in organisms [99]. Thus, enhanced HDL and LDL levels along with reduced levels of triglyceride detected in the probiotic-/synbiotic-fed groups in the present study might indicate an improved lipid profile and better health status in fish compared to the control group. A significant increase in the plasma cholesterol levels in rainbow trout, *Oncorhynchus mykiss*, fed a probiotic-supplemented diet has been reported [100]. Conversely, a decrease in triglyceride levels owing to feeding diets supplemented with pre- and/or probiotics was recorded for Japanese flounder, *Paralichthys olivaceus* [81]. Probiotic and synbiotic diets might regulate the cholesterol level by converting it into coprostanol [101]. The present study indicated a decrease in serum glucose levels in fish that received probiotics/synbiotics through their diets. Our study contradicts some previous reports representing increased blood glucose levels in grass carp, *C. idella* [97], and *L. rohita* [2] fed dietary synbiotic- and/or probiotic-supplemented diets. As suggested, certain immunostimulants could lower serum glucose levels by increasing the level of insulin [102]. Thus, the decrease in the serum glucose level observed in the present study might suggest improved glucose utilisation and good health status in fish fed functional feed additives. The decline in serum SGPT and SGOT levels detected in the present study in the experimental fish as compared to the control group might suggest that the prebiotic/probiotics/synbiotics used were safe for the metabolic health of the fish [2,5]. Even though our result was inconsistent with Park et al. [103] in terms of SGOT, SGPT and glucose levels remained more or less unchanged in starry flounder, *Platichthys stellatus*, fed probiotic-supplemented diets. Although serum LDH levels were almost unaffected due to the dietary administration of functional feed additives, challenge with pathogenic *A. hydrophila* elevated LDH levels in all experimental groups, indicating tissue damage in the exposed fish. LDH is usually released during tissue damage, and thus, increased LDH might suggest common infections or injuries in organisms [104].

As a whole, analyses of haematological as well as serum biochemical parameters might suggest that groups fed probiotics and synbiotics (*B. licheniformis +* FOS in particular) were associated with better health status than the control or only FOS-supplemented groups. The haemato-biochemical parameters recorded in the present study were more or less consistent with the previous reports, while species differences, strain variation and environmental factors might be responsible for minor variations [5,105].

### 4.4. Immune Parameters

The serum consists of numerous peptides, e.g., lysozymes, antiprotease, peroxidase, complement and lytic components, representing innate or non-specific immune responses that provide the first line of immune defence and prevent the adherence as well as colonisation of pathogens [106]. Lysozyme is an important indicator of the non-specific immune response, the activity of which might be increased in fish against bacterial, viral and parasitic infections [21]. Becteriolysis caused by lysozyme could influence the complement along with phagocytic activity [107]. In the present study, probiotic- and synbiotic-fed groups experienced significant improvement (*p* < 0.05) in serum and mucus lysozyme activity, which was the highest in the fish fed diet E4. Elevated lysozyme levels notably coincided with increased leukocyte counts, exhibiting immune-enhancing properties of the pro- and synbiotics. The increase in serum lysozyme activity in fish owing to the administration of dietary prebiotics [108,109] and probiotics [2,5,9] has been documented in previous studies. In accordance with the elevated serum lysozyme activity observed in the present work, the application of dietary synbiotics consisting of FOS and *Bacillus* spp. could raise the serum lysozyme activity in Japanese flounder, *Paralichthys olivaceus* [81]; juvenile yellow croaker, *Larimichthys crocea* [11]; juvenile ovate pompano, *Trachinotus ovatus* [70]; and *L. rohita* juveniles [21]. For example, a dietary synbiotic consisting of *B. circulans* PB7 and FOS could improve the growth and immune-physiological function in *L. rohita* juveniles exposed to low pH stress [21]. Similarly, the application of *P. acidilactici* + GOS improved the serum lysozyme activity in rainbow trout, *Oncorhynchus mykiss* [6]. In contrast, the application of a commercial synbiotic formulation (*Biomin*^®^*IMBO)* containing *Entercoccus faecium* (5 × 10^11^ CFU kg^−1^) and FOS could not significantly improve serum lysozyme levels in beluga, *Huso huso*, after 8 weeks of trials [110]. Skin mucus, by secreting various biostatic and biocidal molecules, serves as a major biological barrier for different pathogens [111]. A previous report by Modanloo et al. [112] indicated a likely defensive role of the skin mucus lysozyme against some invasive bacterial infections. Our study observed a significant improvement in the skin mucus lysozyme activity in *L. rohita* fed a dietary synbiotic containing *B. licheniformis +* FOS, which was consistent with the observation by Hoseinifar et al. [6] documenting improved skin mucus activity in rainbow trout, *Oncorhynchus mykiss*, fed *P. acidilactici* with GOS as a synbiotic. Complements are major soluble glycoproteins that enhance the ability of antibodies and phagocytic cells to remove microbes and damaged cells from an organism through opsonisation [113]. In the present study, significantly enhanced ACP activities were recorded in *L. rohita* fingerlings fed dietary synbiotics and probiotics, although the best result was achieved with the synbiotic diet E4 (*B. licheniformis +* FOS). Similar to our observation, Caspian roach fed a synbiotic diet with *Entercoccus faecium* and FOS (2 g kg^−1^) revealed increased levels of lysozyme and ACH_50_ [114]. In accordance with our finding, significantly increased levels of serum lysozyme and ACP were recorded in snow trout fed diets enriched with FOS (10–30 g kg^−1^) [115]. Elevated serum ACH_50_ was also noticed in common carp, *Cyprinus carpio*, fed dietary GOS and *Pediococcus acidilactici* [112].

Peroxidase enzyme produces hypochlorous acid that oxidises radicals to kill the pathogen. Our study revealed that both serum and mucus peroxidase activities were significantly higher in fish fed synbiotic and probiotic supplements, while minor improvement was observed in the group supplemented with only FOS. Enhanced serum peroxidase activity due to feeding probiotic-supplemented diets was recorded in several previous reports, e.g., in Nile tilapia [116] and *L. rohita* [2,5]. Further, in accordance with the present report, the use of synbiotic in the form of corncob-derived xylooligosaccharide and *Lactobacillus plantarum* CR1T5 could enhance the serum peroxidase activity in Nile Tilapia, *Oreochromis niloticus*, fingerlings [117]. For skin mucus peroxidase, the administration of probiotic beta-glucan promoted mucus peroxidase in red sea bream, *Pagrus major* [118]. The combined application of *Cordyceps militaris* spent mushroom substrate and *L. plantarum* in diets induced skin mucus peroxidase in Nile tilapia, *Oreochromis niloticus*, in an 8-week experiment [12]. Anti-protease activities of serum are mainly due to α1- and α2-antiproteases along with α2-macroglobulin that hinder the activities of the proteases produced by the opportunistic bacteria to attack the host. Feeding synbiotic and probiotic supplements could improve anti-protease activity in *L. rohita* fingerlings, as evidenced in the present study. Although the effects of synbiotics on serum anti-protease activity in fish are rarely addressed, probiotics-induced improvements in the anti-protease activity in fish were documented in *O. mykiss* [119] and *L. rohita* [2].

Phagocytic activity represents the early activation of the innate immune response exhibited by the cell-mediated killing of the aggressive pathogens attacked by respiratory bursts activity associated with the production of free radicals that somehow clear foreign particles [33]. Thus, phagocytosis and respiratory bursts of the head kidney leucocytes have been considered as an index to assess defence ability against pathogens [2]. In our study, the administration of dietary synbiotics and probiotics significantly improved phagocytosis before and after the pathogen challenge study. Like our observation, a combination of *Bacillus subtilis* and FOS could improve the phagocytic activity in *Trachinotus ovatus* [70]. In a previous record, the incorporation of FOS and MOS (each at 2.5 g kg^−1^) along with *B. clausii* (10^7^ cells g^−1^) in diets enhanced the phagocytic percentage in Japanese flounder, *Paralichthys olivaceus* [81]. Apart from synbiotics, probiotic-induced improvements in phagocytic activity were documented in several studies, e.g., *Epinephelus coioides* fed diets with *B. subtilis* E20 (10^4^, 10^6^ and 10^8^ CFU g^−1^) [120] and *L. rohita* fed diets containing bacilli [2,121]. In addition, augmentation in phagocytosis was also recorded in *Catla catla* fed diets with *B. circulans* and challenged with *A. hydrophila* [122]. Respiratory bursts (oxidative stress) are the rapid production of reactive oxygen species (ROS) and reactive nitrogen species (RNS) by the head kidney leucocytes, neutrophils and macrophage cells to kill internalised pathogens by phagocytosis. The three primary species, i.e., the superoxide anion (O_2_^•−^), the hydroxyl radical (HO^•^) and hydrogen peroxide (H_2_O_2_), are known as ROSs, which are produced to kill the internal pathogen and promote the inflammatory process [123,124]. In the presently reported study, fish fed dietary synbiotics and probiotics were recorded with the increase in respiratory burst activity and super oxide anion production, which were further augmented after the pathogen challenge. This probably entails a better protective role of the probiotic and synbiotic supplements associated with an improved bacterial-pathogen-destroying potential of the phagocytes with respiratory bursts of activity and might be correlated with elevated lysozyme levels, as noticed in the present study [21]. Our findings were consistent with Devi et al. [3], in which single or combined applications of *Bacillus subtilis* and GOS were reported to promote respiratory bursts, as well as serum lysozyme activities in *L. rohita*, even when challenged against *A. hydrophila*. In addition, diverse fish species fed different dietary probiotic supplements were reported with similar observations [2,29,55,125]. Apart from probiotics or synbiotics, feeding FOS (2%)-enriched diets could significantly enhance the respiratory burst and serum lysozyme activities in climbing perch, *Aanabas testudineus* [126]. Conversely, no significant difference was detected in the present study for the respiratory bursts of activity before and after the pathogen challenge, when only FOS was used as a feed supplement for *L. rohita* fingerlings. Considering natural prebiotics, diets containing garlic (10 g kg^−1^) were observed to stimulate superoxide anion production in *Labeo rohita* [127]. Overall, the synergistic activity of probiotic and prebiotic supplements as synbiotics might be considered of great importance for immune modulation in fish, as evidenced in the present report.

### 4.5. Stress Parameters

Oxidative stress is a cellular phenomenon resulting from the imbalance between free radicals and some metabolic preservatives that make unstable free radicals change to a stable form and help to eliminate their toxicity [128]. The free radicals produced in the stressed condition are removed by antioxidant enzymes. Lipid peroxidation is considered a biomarker for oxidative stress in fish [129]. The assessment of SOD and catalase may provide important tools to assess the health condition of hosts [130,131]. In our study, SOD and catalase activities were enhanced in fish that received synbiotic-supplemented diets. Similar antioxidant potential was observed in *L. rohita* fingerlings fed diets supplemented with probiotics [132] or synbiotics [3]. Further, in agreement with our study, feeding diets with *B. subtilis* and MOS (0.6%) could increase SOD and catalase activities in mrigal carp, *Crihinus mrigala* [124]. In another study, the *administration of* β-glucan + *Pediococcus acidilactici could improve SOD activity in pacific white shrimp compared to the basal diet* [133]. Malondialdehyde (MDA) is one of the final products of the peroxidation of polyunsaturated fatty acids in cells. Increasing MDA content might lead to the overproduction of free radicals, which in turn damages the membrane structure [132,134]. Our study revealed lower MDA concentration in the synbiotic-fed groups, indicating stronger anti-oxidant profiles in the experimental fish. Antioxidant enzymes enhance the immune profile by removing free radicals and also by lowering the lipid damage by the radicals [132]. It is likely that, in our study, SOD and catalase activities inversely correlated with MDA concentration, resulting in improved antioxidant potential. Consistent with our observation, the improved antioxidant health status in Asian sea bass, *Lates calcarifer*, juveniles fed synbiotic diets was recorded in Ashouri et al. [78].

### 4.6. Challenge Study

The effectiveness of dietary treatment in terms of pathogen prevention and survival percentage could be demonstrated with a challenge study involving pathogen markers [135]. It has been hypothesised that the application of synbiotics might ensure disease resistance in aquaculture by improving the function of probiotics [6]. *Aeromonas hydrophila* is one of the major pathogenic bacteria that hamper fish health in tropical countries [2]. The antagonism against fish-pathogenic *Aeromonas* spp. and the production of antibacterial substances by the probiotic strains used in the present report were documented in previous studies [2,31]. Several previous studies recorded the effectiveness of the dietary administration of probiotics to improve disease resistance in fish against aeromonad infections [2,29,136]. Consistent with these observations, the present study also demonstrated significant improvement in the survivability of fish exposed to *A. hydrophila* following the application of dietary probiotics (diets E2 and E3). However, the survivability increased when probiotics were used in combination with FOS as synbiotics.

The maximum survivability (%) of fish fed diet E4 might be correlated with the modulation of immune parameters in *L*. *rohita* fed synbiotics containing *B. licheniformis* and FOS. Compatible with the present report, supplementation of *P. acidilactici* and GOS in the diet provided better survivability in rainbow trout, *Oncorhynchus mykiss*, challenged with *Streptococcus iniae* [6]. Further, *B. subtilis* fed with FOS displayed a significantly lower mortality rate in juvenile large yellow croaker, *Larimichthys crocea*, after infection with pathogenic *V.harveyi* [11].

## 5. Conclusions

The results of the presently reported study clearly indicate that the dietary administration of *B. licheniformis* or *B. methylotrophicus* along with FOS improved not only feed utilisation and growth performance in *L. rohita* fingerlings but also the non-specific immune function and resistance against *A. hydrophila* infection. The statistical analyses with PCA and Pearson’s correlation matrix plot showed a strong positive association of digestive enzymes, serum biochemical parameters (total protein, globulin, alkaline phosphatase) and stress enzymes (superoxide dismutase, catalase), establishing the fact that dietary probiotics and synbiotics might support better growth, feed utilisation and immunity in fish. FOS or probiotic bacilli alone could also produce significant improvement in the performance of the experimental fish compared to the control group, while application in the form of synbiotics created much better results. Prophylactic application of the synbiotics appeared to protect against pathogenic exposure to a great extent and considerably improved the survivability of pathogen-challenged fish. The presently reported study used FOS as the prebiotic supplement at the 1% level. However, further studies pertaining to the single or combined applications of diverse probiotic strains in the formulation of synbiotics along with appropriate levels of incorporation of prebiotics remain to be carried out. This study indicated that the synbiotic diets brought about better antioxidant and stress-management potential when compared to the sole applications of prebiotics or probiotics. However, innate and adaptive immune responses along with gut immunity in fish with reference to the synbiotics application should be investigated in forthcoming studies. Detailed investigations on the expression of growth- and immunity-related genes might be targeted to realise the underlying mechanisms of the synbiotic functions. Overall, this study might offer an alternative avenue for the management of bacterial infections in fish to secure the enhanced production of carp through environmentally friendly aquaculture. Even so, further studies involving field trials are required before the synbiotic composition can be recommended for commercial aquaculture.

## Figures and Tables

**Figure 1 animals-13-02631-f001:**
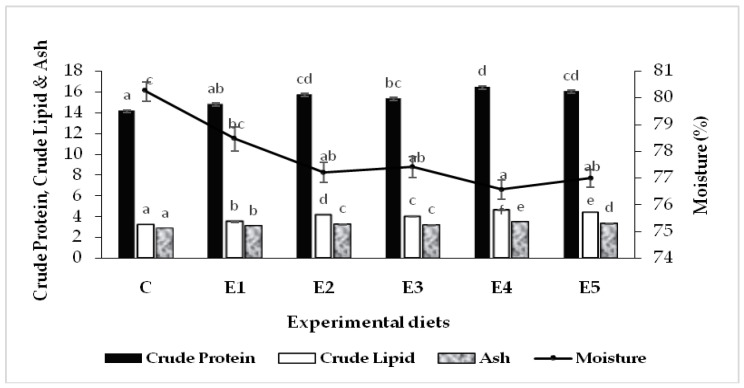
Proximate carcass composition (% wet weight) of *L. rohita* fingerlings at the end of the 90-day feeding trial. Values are means ± SE (*n* = 3). Similar lower-case letters for a parameter are not significantly different (*p* < 0.05).

**Figure 2 animals-13-02631-f002:**
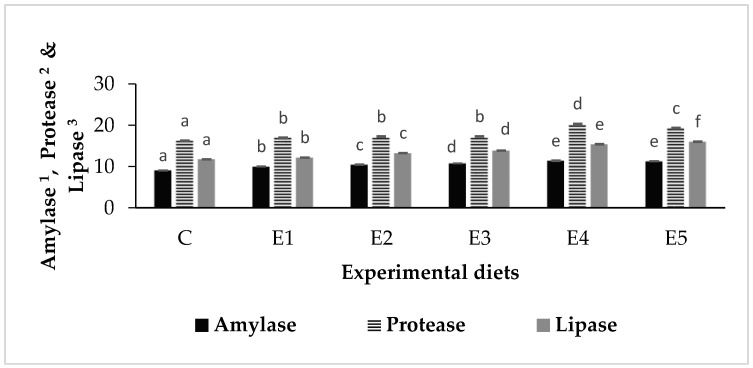
Activities of the digestive enzymes in *L. rohita* fingerlings fed diets containing different functional feed additives for 90 days. Values are means ± SE (*n* = 3). Similar lower-case letters for a parameter are not significantly different (*p* < 0.05). ^1^ mg of maltose liberated mg^−1^ protein h^−1^; ^2^ μg of tyrosine liberated mg^−1^ protein h^−1^; ^3^ μmole of fatty acid liberated mg^−1^ protein h^−1^.

**Figure 3 animals-13-02631-f003:**
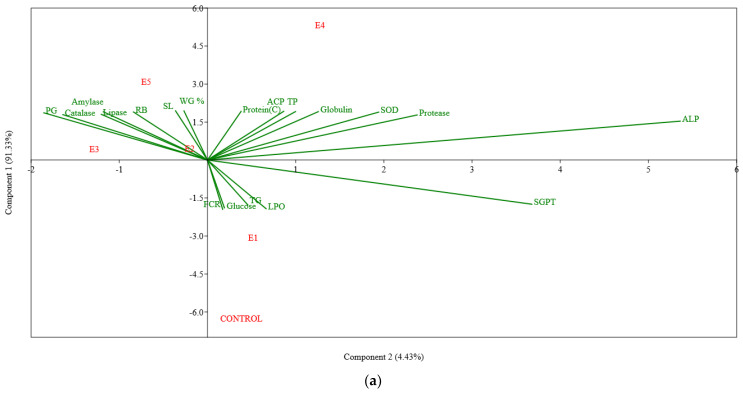
Ordination diagram of PCA plot (**a**) and Pearson correlation matrix plot (**b**) with weight gain (%), carcass protein, digestive enzymes, serum biochemical and immune and stress parameters in *Labeo rohita* fingerlings fed experimental diets for 90 days. In (**b**), the blue represents a positive correlation and red represents a negative correlation. The box represents that the correlation was significant (*p* < 0.05). Abbreviations: WG%, Weight Gain %; FCR, Feed Conversion Ratio; Protein (C), Carcass Protein; TP, Total Serum Protein; TG, Triglyceride; ALP, Alkaline Phosphatase; SGPT, Serum Glutamic Pyruvic Transaminase; SL, Serum Lysozyme; ACP, Alternative Complement Pathway; PG, Phagocytic Activity; RB, Respiratory Burst; SOD, Superoxide dismutase; LPO, Lipid Peroxidation.

**Figure 4 animals-13-02631-f004:**
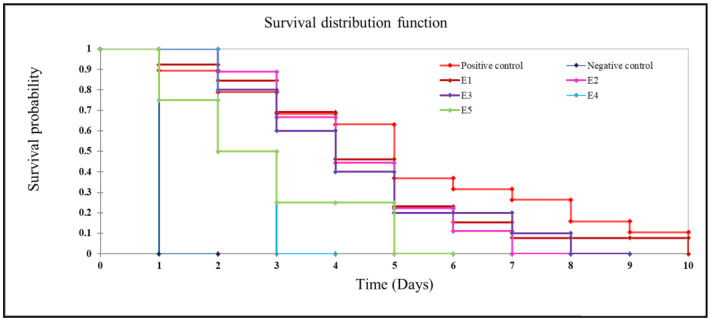
Kaplan–Meier survival plots of *Labeo rohita* after challenge with *A. hydrophila*.

**Table 1 animals-13-02631-t001:** Ingredient profile (in dry weight) and proximate composition of the basal diet (on % dry matter).

Ingredient Profile (on % Dry Weight)		Proximate Composition	
Fish meal	37	Dry matter	92.32 ± 0.49
Mustard oil cake	25	Crude protein	35.27 ± 0.20
Rice bran	15	Crude lipid	6.71 ± 0.03
Wheat bran	20	Crude fibre	13.66 ± 0.14
Cod liver oil	1	Nitrogen-free extract	21.49 ± 0.12
Vitamin + mineral premix ^a^	1	Ash	14.69 ± 0.07
Carboxymethylcellulose	1	Gross energy (MJkg^−1^)	17.06 ± 0.09

Values are mean ± SE (*n* = 3). ^a^ Supradyn, Bayer Consumer Care AG.

**Table 2 animals-13-02631-t002:** Feeding schedule of the experimental groups.

Experimental Groups	Diets Used	FOS	*B. licheniformis* (KU556167)	*B. methylotrophicus* (KU556164)
C	Basal	-	-	-
E1	Basal + Prebiotic	1%	-	-
E2	Basal + Probiotic	-	7 Log CFU/g	-
E3	Basal + Probiotic	-	-	7 Log CFU/g
E4	Basal + Synbiotic	1%	7 Log CFU/g	-
E5	Basal + Synbiotic	1%	-	7 Log CFU/g

**Table 3 animals-13-02631-t003:** Growth performance and feed utilisation efficacy in *Labeo rohita* fingerlings fed experimental diets for 90 days.

Parameters	C	E1	E2	E3	E4	E5
Initial weight (g)	10.21 ± 0.17	10.15 ± 0.15	10.1 ± 0.17	10.17 ± 0.16	10.19 ± 0.17	10.25 ± 0.16
Final weight (g)	29.97 ± 0.31 ^a^	31.64 ± 0.29 ^b^	33.87 ± 0.32 ^c^	34.07 ± 0.32 ^c^	36.66 ± 0.40 ^d^	35.39 ± 0.25 ^c^
Weight gain (%)	193.53 ± 2.03 ^a^	211.72 ± 1.92 ^b^	235.34 ± 2.26 ^c^	234.7 ± 2.21 ^c^	259.72 ± 2.83 ^d^	245.27 ± 1.77 ^c^
SGR (% day^−1^)	1.19 ± 0.02 ^a^	1.26 ± 0.01 ^a^	1.34 ± 0.02 ^b^	1.34 ± 0.01 ^b^	1.42 ± 0.02 ^c^	1.37 ± 0.01 ^bc^
FCR ^1^	2.4 ± 0.01 ^e^	2.22 ± 0.01 ^d^	2.0 ± 0.01 ^c^	2.02 ± 0.02 ^c^	1.74 ± 0.01 ^a^	1.84 ± 0.01 ^b^
PER ^2^	1.19 ± 0.02 ^a^	1.29 ± 0.02 ^b^	1.43 ± 0.01 ^c^	1.41 ± 0.01 ^c^	1.64 ± 0.02 ^e^	1.55 ± 0.02 ^d^
ANPU ^3^ (%)	17.98 ± 0.27 ^a^	20.57 ± 0.33 ^b^	24.53 ± 0.34 ^c^	23.47 ± 0.25 ^c^	29.58 ± 0.31 ^e^	27.19 ± 0.40 ^d^
Survivability (%)	93.33 ± 1.92	96.66 ± 0.00	96.66 ± 1.92	93.33 ± 3.9	96.66 ± 1.92	96.66 ± 0.00
Condition factor (K)	0.92 ± 0.01 ^a^	1.1 ± 0.01 ^b^	1.2 ± 0.01 ^c^	1.16 ± 0.01 ^c^	1.52 ± 0.01 ^e^	1.43 ± 0.01 ^d^

Values are means ± SE (*n* = 3). Values with the same superscripts in the same row are not significantly different (*p* < 0.05). ^1^ Feed conversion ratio; ^2^ Protein efficiency ratio; ^3^ Apparent net protein utilization.

**Table 4 animals-13-02631-t004:** Haematological parameters in rohu fingerlings fed diets containing pre-, pro- or synbiotics for 90 days and after the challenge study with *A. hydrophila*.

After 90 Days						
Parameters	C	E1	E2	E3	E4	E5
RBC (×10^6^/mm^3^)	2.77 ± 0.01 ^a^	2.77 ± 0.02 ^a^	2.85 ± 0.02 ^ab^	2.78 ± 0.01 ^ab^	3.05 ± 0.02 ^c^	2.86 ± 0.02 ^b^
WBC (×10^2^/mm^3^)	180 ± 0.88 ^a^	198 ± 0.93 ^b^	204 ± 0.97 ^c^	212 ± 1.02 ^d^	221 ± 0.97 ^e^	221 ± 1.24 ^e^
Hb (g/dL)	10.7 ± 0.11 ^ab^	10.4 ± 0.13 ^a^	11.3 ± 0.12 ^bc^	11.0 ± 0.14 ^ab^	11.8 ± 0.16 ^c^	11.0 ± 0.11 ^ab^
PCV (%)	40.0 ± 0.21 ^a^	40.1 ± 0.21 ^a^	43.3 ± 0.19 ^c^	43.3 ± 0.22 ^c^	45.6 ± 0.22 ^d^	41.9 ± 0.18 ^b^
ESR (mm/h)	3.8 ± 0.02 ^c^	3.3 ± 0.02 ^b^	2.6 ± 0.02 ^a^	2.5 ± 0.01 ^a^	2.6 ± 0.02 ^a^	2.5 ± 0.01 ^a^
Platelet (×10^3^/mm^3^)	39 ± 0.42 ^b^	36 ± 0.75 ^a^	60 ± 0.73 ^f^	45 ± 0.67 ^d^	56 ± 0.19 ^e^	42 ± 0.16 ^c^
MCV(μm^3^)	144.4 ± 0.76 ^ab^	141.15 ± 0.71 ^a^	151.9 ± 0.83 ^d^	155.8 ± 0.93 ^e^	149.5 ± 0.79 ^cd^	146.5 ± 0.72 ^bc^
MCH (pg/cell)	38.62 ± 0.21 ^bc^	37.54 ± 0.18 ^a^	39.6 ± 0.20 ^c^	39.6 ± 0.18 ^c^	38.7 ± 0.24 ^bc^	38.5 ± 0.30 ^ab^
MCHC (g%)	26.8 ± 0.11 ^c^	25.9 ± 0.13 ^ab^	26.1 ± 0.13 ^b^	25.4 ± 0.15 ^a^	25.9 ± 0.15 ^ab^	26.3 ± 0.13 ^bc^
**After challenge with *Aeromonas hydrophila***						
RBC (×10^6^/mm^3^)	1.97 ± 0.01 ^bc^	1.86 ± 0.01 ^a^	1.92 ± 0.01 ^b^	1.98 ± 0.01 ^c^	2.12 ± 0.01 ^d^	2.02 ± 0.01 ^c^
WBC (×10^2^/mm^3^)	252 ± 1.43 ^a^	254 ± 1.23 ^ab^	262 ± 1.33 ^cd^	258 ± 1.28 ^bc^	264 ± 1.20 ^d^	257 ± 0.97 ^ab^
Hb (g/dL)	7.2 ± 0.07 ^a^	7.9 ± 0.09 ^b^	8.1 ± 0.09 ^b^	7.9 ± 0.08 ^b^	8.9 ± 0.11 ^c^	8.0 ± 0.10 ^b^
PCV (%)	33.3 ± 0.14 ^a^	33.1 ± 0.20 ^a^	34.5 ± 0.15 ^bc^	33.5 ± 0.18 ^ab^	36.5 ± 0.20 ^d^	34.8 ± 0.21 ^c^
ESR (mm/h)	4.7 ± 0.02 ^d^	3.7 ± 0.02 ^b^	3.8 ± 0.02 ^c^	3.8 ± 0.02 ^c^	3.3 ± 0.02 ^a^	3.7 ± 0.02 ^b^
Platelet (×10^3^/mm^3^)	35 ± 0.43 ^a^	37 ± 0.20 ^b^	52 ± 0.40 ^f^	43 ± 0.31 ^d^	50 ± 0.35 ^e^	41 ± 0.23 ^c^
MCV (μm^3^)	169.13 ± 1.09 ^a^	178.06 ± 0.95 ^b^	179.89 ± 0.82 ^b^	169.39 ± 0.96 ^a^	171.93 ± 0.82 ^a^	172.07 ± 0.87 ^a^
MCH (pg/cell)	36.59 ± 0.30 ^a^	42.47 ± 0.16 ^c^	42.29 ± 0.40 ^c^	39.74 ± 0.36 ^b^	41.83 ± 0.37 ^c^	39.7 ± 0.29 ^b^
MCHC (g%)	21.63 ± 0.11 ^a^	23.85 ± 0.10 ^cd^	23.5 ± 0.13 ^bc^	23.46 ± 0.13 ^bc^	24.33 ± 0.14 ^d^	23.07 ± 0.12 ^b^

Values are means ± SE (*n* = 3). Values with the same superscripts in the same row are not significantly different (*p* < 0.05). Abbreviations: RBC, total red blood cells count; WBC, total white blood cells count; Hb, haemoglobin; PCV, Packed Cell Volume; ESR, Erythrocyte Sedimentation Rate; MCV, Mean Corpuscular Volume; MCH, Mean Corpuscular Haemoglobin; MCHC, Mean Corpuscular Haemoglobin Concentration. Initial values (day 0): RBC, 2.68 ± 0.03; WBC, 176 ± 0.80; Hb, 8.6 ± 0.09; PCV, 37.5 ± 0.22; Platelet, 34 ± 0.17; MCV,139.92 ± 0.90; MCH, 32.08 ± 0.21; MCHC, 22.8 ± 0.12.

**Table 5 animals-13-02631-t005:** Serum-biochemical parameters in rohu fingerlings fed diets supplemented with pre-, pro- or synbiotics for 90 days and after the challenge study with *A. hydrophila*.

After 90 Days						
Parameters	C	E1	E2	E3	E4	E5
Total protein (g dL^−1^)	2.71 ± 0.02 ^a^	2.86 ± 0.02 ^b^	3.12 ± 0.01 ^c^	3.07 ± 0.03 ^c^	3.41 ± 0.02 ^d^	3.16 ± 0.02 ^c^
Albumin (g dL^−1^)	1.30 ± 0.01 ^a^	1.37 ± 0.01 ^b^	1.51 ± 0.01 ^c^	1.50 ± 0.004 ^c^	1.66 ± 0.01 ^d^	1.53 ± 0.01 ^c^
Globulin (g dL^−1^)	1.41 ± 0.02 ^a^	1.49 ± 0.01 ^b^	1.61 ± 0.02 ^c^	1.57 ± 0.02 ^c^	1.75 ± 0.01 ^d^	1.63 ± 0.01 ^c^
Cholesterol (mg dL^−1^)	128 ± 2.12 ^a^	133 ± 2.01 ^a^	145 ± 1.36 ^b^	142 ± 0.46 ^b^	150 ± 2.23 ^bc^	155 ± 1.49 ^c^
HDL (mg dL^−1^)	31 ± 0.33 ^a^	50 ± 0.18 ^b^	55 ± 0.26 ^c^	55 ± 0.18 ^c^	60 ± 0.20 ^d^	61 ± 0.23 ^d^
LDL (mg dL^−1^)	78 ± 0.84 ^b^	72 ± 0.26 ^a^	81 ± 0.38 ^c^	78 ± 0.25 ^b^	84 ± 0.28 ^d^	85 ± 0.32 ^d^
Triglyceride (mg dL^−1^)	95 ± 0.47 ^d^	55 ± 0.35 ^c^	45 ± 0.25 ^b^	45 ± 0.22 ^b^	30 ± 0.23 ^a^	45 ± 0.29 ^b^
Glucose (mg dL^−1^)	127.4 ± 0.63 ^d^	125.5 ± 0.60 ^cd^	121.5 ± 0.68 ^ab^	123 ± 0.78 ^bc^	118 ± 0.89 ^a^	118 ± 1.19 ^a^
ALP (µ cat L^−1^)	0.55 ± 0.002 ^a^	0.73 ± 0.002 ^c^	0.72 ± 0.002 ^c^	0.62 ± 0.002 ^b^	1.15 ± 0.004 ^e^	0.77 ± 0.003 ^d^
SGPT (µ cat L^−1^)	1.85 ± 0.02 ^c^	1.72 ± 0.02 ^b^	1.59 ± 0.02 ^a^	1.55 ± 0.02 ^a^	1.57 ± 0.02 ^a^	1.52 ± 0.02 ^a^
SGOT (µ cat L^−1^)	1.46 ± 0.03 ^b^	1.45 ± 0.02 ^b^	1.23 ± 0.02 ^a^	1.41 ± 0.02 ^b^	1.22 ± 0.01 ^a^	1.39 ± 0.01 ^b^
LDH (µ cat L^−1^)	6.26 ± 0.07 ^b^	6.24 ± 0.02 ^b^	6.04 ± 0.03 ^a^	6.1 ± 0.02 ^ab^	6.01 ± 0.02 ^a^	6.12 ± 0.02 ^ab^
**After challenge with *Aeromonas hydrophila***						
Total protein (g dL^−1^)	2.61 ± 0.01 ^a^	2.78 ± 0.03 ^b^	3.27 ± 0.03 ^c^	3.35 ± 0.02 ^c^	3.58 ± 0.02 ^d^	3.6 ± 0.02 ^d^
Albumin (g dL^−1^)	1.18 ± 0.01 ^a^	1.24 ± 0.01 ^b^	1.43 ± 0.01 ^c^	1.42 ± 0.01 ^c^	1.58 ± 0.01 ^e^	1.52 ± 0.01 ^d^
Globulin (g dL^−1^)	1.43 ± 0.02 ^a^	1.54 ± 0.01 ^b^	1.84 ± 0.01 ^c^	1.93 ± 0.01 ^d^	2.00 ± 0.01 ^e^	2.08 ± 0.01 ^f^
Cholesterol (mg dL^−1^)	175 ± 0.76 ^c^	178 ± 0.84 ^c^	160 ± 1.71 ^a^	161 ± 1.69 ^ab^	165 ± 0.84 ^ab^	167 ± 1.08 ^b^
HDL (mg dL^−1^)	34 ± 0.19 ^a^	48 ± 0.26 ^b^	50 ± 0.25 ^c^	53 ± 0.31 ^e^	51 ± 0.26 ^cd^	52 ± 0.34 ^de^
LDL (mg dL^−1^)	115 ± 0.63 ^d^	106 ± 0.58 ^c^	95 ± 0.47 ^a^	93 ± 0.55 ^a^	98 ± 0.50 ^b^	98 ± 0.63 ^b^
Triglyceride (mg dL^−1^)	130 ± 0.74 ^e^	120 ± 0.93 ^d^	75 ± 0.38 ^a^	75 ± 0.37 ^a^	80 ± 0.50 ^b^	85 ± 0.54 ^c^
Glucose (mg dL^−1^)	142.7 ± 0.81 ^c^	141.4 ± 1.09 ^c^	134.86 ± 0.67 ^ab^	138.85 ± 0.69 ^bc^	134.1 ± 0.84 ^a^	138.76 ± 0.89 ^bc^
ALP(µ cat L^−1^)	0.87 ± 0.003 ^c^	0.85 ± 0.004 ^b^	0.87 ± 0.004 ^c^	0.79 ± 0.005 ^a^	1.23 ± 0.005 ^d^	0.86 ± 0.004 ^bc^
SGPT (µ cat L^−1^)	2.11 ± 0.02 ^c^	2.05 ± 0.02 ^c^	1.89 ± 0.02 ^b^	1.87 ± 0.01 ^b^	1.76 ± 0.02 ^a^	1.81 ± 0.02 ^ab^
SGOT (µ cat L^−1^)	1.87 ± 0.02 ^c^	1.84 ± 0.01 ^c^	1.36 ± 0.01 ^a^	1.49 ± 0.01 ^b^	1.34 ± 0.01 ^a^	1.44 ± 0.01 ^b^
LDH (µ cat L^−1^)	8.78 ± 0.05 ^b^	8.72 ± 0.05 ^b^	8.45 ± 0.05 ^a^	8.44 ± 0.04 ^a^	8.41 ± 0.05 ^a^	8.32 ± 0.05 ^a^

Values are means ± SE (*n* = 3). Values with same superscripts in the same row are not significantly different (*p* < 0.05). Abbreviations: HDL, High-Density Lipoprotein; LDL, Low-Density Lipoprotein; ALP, Alkaline Phosphatase; SGPT, Serum Glutamic Pyruvic Transaminase; SGOT, Serum Glutamic-Oxaloacetic Transaminase; LDH, Lactate Dehydrogenase. Initial values (day 0): Total protein, 2.65 ± 0.02; Albumin, 1.27 ± 0.01; Globulin, 1.38 ± 0.01; Cholesterol, 125 ± 0.80; HDL, 31 ± 0.16; LDL, 75 ± 0.38; Triglyceride, 95 ± 0.97; Glucose, 128.3 ± 0.81; ALP, 1.22 ± 0.004; SGPT, 1.89 ± 0.03; SGOT, 1.43 ± 0.01; LDH, 5.65 ± 0.03.

**Table 6 animals-13-02631-t006:** Immune parameters in rohu fingerlings fed diets supplemented with pre-, pro- or synbiotics for 90 days and after the challenge study with *A. hydrophila*.

After 90 Days						
Parameters	C	E1	E2	E3	E4	E5
Serum Lysozyme ^1^	14.52 ± 0.09 ^a^	16.42 ± 0.14 ^b^	18.78 ± 0.10 ^c^	18.75 ± 0.14 ^c^	21.23 ± 0.15 ^e^	19.87 ± 0.13 ^d^
Mucus Lysozyme ^2^	3.03 ± 0.05 ^a^	3.12 ± 0.02 ^a^	3.93 ± 0.07 ^b^	3.97 ± 0.07 ^b^	5.05 ± 0.09 ^c^	4.1 ± 0.07 ^b^
ACP ^3^	127.09 ± 0.82 ^a^	152.45 ± 1.26 ^b^	166.69 ± 0.90 ^c^	162 ± 1.20 ^c^	194.03 ± 1.41 ^e^	178.23 ± 1.19 ^d^
Serum Peroxidase ^4^	0.52 ± 0.012 ^a^	0.68 ± 0.016 ^b^	0.94 ± 0.022 ^c^	1.05 ± 0.023 ^d^	1.13 ± 0.020 ^de^	1.17 ± 0.032 ^e^
Mucus Peroxidase ^5^	0.13 ± 0.004 ^a^	0.19 ± 0.005 ^b^	0.26 ± 0.007 ^c^	0.29 ± 0.008 ^c^	0.35 ± 0.009 ^d^	0.37 ± 0.010 ^d^
Antiprotease ^6^	72.71 ± 0.53 ^a^	72.96 ± 0.39 ^a^	78.36 ± 0.31 ^c^	76.17 ± 0.87 ^b^	80.98 ± 0.45 ^d^	84.27 ± 0.42 ^e^
Phagocytosis ^7^	23.43 ± 0.25 ^a^	24.76 ± 0.52 ^a^	31.84 ± 0.39 ^b^	35.23 ± 0.52 ^c^	38.54 ± 0.13 ^d^	36.76 ± 0.14 ^c^
Super oxide anion ^8^	38.12 ± 0.69 ^a^	39.65 ± 0.10 ^a^	43.23 ± 0.78 ^bc^	41.2 ± 0.74 ^ab^	46.54 ± 0.84 ^c^	45.65 ± 0.82 ^c^
RB ^9^	0.22 ± 0.005 ^a^	0.23 ± 0.008 ^a^	0.34 ± 0.009 ^b^	0.34 ± 0.010 ^b^	0.42 ± 0.012 ^d^	0.39 ± 0.010 ^c^
**After challenge with *Aeromonas hydrophila***						
Serum Lysozyme ^1^	17.45 ± 0.09 ^a^	19.85 ± 0.11 ^b^	22.63 ± 0.09 ^c^	23.57 ± 0.31 ^d^	25.77 ± 0.12 ^f^	24.61 ± 0.30 ^e^
Mucus Lysozyme ^2^	3.84 ± 0.07 ^a^	3.95 ± 0.07 ^a^	4.16 ± 0.02 ^b^	4.29 ± 0.02 ^b^	5.41 ± 0.02 ^c^	4.34 ± 0.02 ^b^
ACP ^3^	166 ± 0.81 ^a^	191.94 ± 1.07 ^b^	196.11 ± 3.12 ^b^	224.22 ± 2.99 ^c^	268.66 ± 1.20 ^e^	236.25 ± 2.37 ^d^
Serum Peroxidase ^4^	0.63 ± 0.014 ^a^	0.82 ± 0.022 ^b^	1.32 ± 0.036 ^c^	1.24 ± 0.029 ^c^	1.46 ± 0.030 ^d^	1.33 ± 0.026 ^c^
Mucus Peroxidase ^5^	0.20 ± 0.005 ^a^	0.29 ± 0.008 ^b^	0.35 ± 0.010 ^c^	0.34 ± 0.008 ^c^	0.43 ± 0.012 ^d^	0.44 ± 0.013 ^d^
Antiprotease ^6^	77.12 ± 0.21 ^a^	79.17 ± 0. 56 ^ab^	82.31 ± 0.74 ^cd^	80.29 ± 0.44 ^bc^	87.37 ± 0.52 ^e^	83.28 ± 0.42 ^d^
Phagocytosis ^7^	28.87 ± 0.36 ^a^	32.07 ± 0.17 ^b^	42.43 ± 0.17 ^d^	40.13 ± 0.19 ^c^	45.87 ± 0.20 ^e^	41.87 ± 0.16 ^d^
Super oxide anion ^8^	40.95 ± 0.74 ^a^	41.78 ± 0.75 ^a^	46.78 ± 0.48 ^b^	46.21 ± 0.28 ^b^	49.52 ± 0.61 ^c^	48.63 ± 0.22 ^bc^
RB ^9^	0.29 ± 0.007 ^a^	0.30 ± 0.007 ^a^	0.41 ± 0.010 ^b^	0.39 ± 0.010 ^b^	0.49 ± 0.012 ^c^	0.48 ± 0.013 ^c^

Values are means ± SE (*n* = 3). Values with same superscripts in the same row are not significantly different (*p* < 0.05). Abbreviations: ACP, Alternative Complement Pathway; RB, Respiratory Burst. ^1^ μg/mL; ^2^ μg/mL; ^3^ units/mL; ^4^ OD 550; ^5^ OD 550; ^6^ trypsin inhibition%; ^7^ %; ^8^ NBT%; ^9^ OD 630. Initial values (day 0): Serum Lysozyme, 5.27 ± 0.04; Mucus Lysozyme, 1.79 ± 0.01; ACP, 121.21 ± 0.92; Serum Peroxidase, 0.35 ± 0.009; Mucus Peroxidase, 0.11 ± 0.003; Antiprotease, 69.74 ± 0.45; Phagocytosis, 18.37 ± 0.09; Super oxide anion, 33.6 ± 0.60; RB, 0.17 ± 0.004.

**Table 7 animals-13-02631-t007:** Stress parameters in rohu fingerlings fed diets supplemented with pre-, pro- or synbiotics for 90 days and after the challenge study with *A. hydrophila*.

After 90 Days						
Parameters	C	E1	E2	E3	E4	E5
SOD ^1^	10.69 ± 0.13 ^a^	12.75 ± 0.19 ^b^	14.38 ± 0.26 ^cd^	13.76 ± 0.28 ^bc^	17.75 ± 0.35 ^e^	15.03 ± 0.24 ^d^
Catalase ^2^	2.97 ± 0.02 ^a^	3.24 ± 0.01 ^b^	3.32 ± 0.02 ^c^	3.8 ± 0.02 ^d^	3.95 ± 0.02 ^e^	3.82 ± 0.02 ^d^
Lipid peroxidation ^3^	19.42 ± 0.23 ^d^	17.68 ± 0.25 ^c^	14.56 ± 0.26 ^b^	15.23 ± 0.31 ^b^	12.76 ± 0.25 ^a^	13.79 ± 0.21 ^ab^
**After challenge with *Aeromonas hydrophila***						
SOD ^1^	14.39 ± 0.20 ^a^	17.48 ± 0.20 ^b^	19.68 ± 0.38 ^c^	18.75 ± 0.31 ^bc^	22.97 ± 0.43 ^d^	19.57 ± 0.28 ^c^
Catalase ^2^	3.23 ± 0.02 ^a^	3.86 ± 0.02 ^c^	3.47 ± 0.02 ^b^	3.95 ± 0.02 ^c^	4.28 ± 0.03 ^e^	4.06 ± 0.02 ^d^
Lipid peroxidation ^3^	23.89 ± 0.33 ^d^	22.04 ± 0.25 ^c^	18.32 ± 0.35 ^b^	19.3 ± 0.32 ^b^	15.76 ± 0.29 ^a^	16.6 ± 0.23 ^a^

Values are means ± SE (*n* = 3). Values with same superscripts in the same row are not significantly different (*p* < 0.05). Abbreviations: SOD, superoxide dismutase. ^1^ U/mg protein; ^2^ μM H_2_O_2_ utilised /mg protein/min; ^3^ nmol/mg protein. Initial values (day 0): SOD, 10.46 ± 0.19; Catalase, 2.2 ± 0.01; Lipid Peroxidation, 20.12 ± 0.36.

**Table 8 animals-13-02631-t008:** Post challenge survival (%) of *Labeo rohita* fingerlings following exposure to *Aeromonas hydrophila*.

Negative Control	Positive Control	E1	E2	E3	E4	E5
96.66 ± 3.33 ^d^	36.66 ± 8.81 ^a^	56.67 ± 3.33 ^ab^	70 ± 5.77 ^bc^	66.67 ± 3.33 ^bc^	93.33 ± 3.33 ^d^	86.66 ± 3.33 ^cd^

Values are means ± SE (*n* = 3). Values with same superscripts in the same row are not significantly different (*p* < 0.05).

## Data Availability

All data used in this study are presented in this article.

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
