# Peer review of "Autochthonous Bacilli and Fructooligosaccharide as Functional Feed Additives Improve Growth, Feed Utilisation, Haemato-Immunological Parameters and Disease Resistance in Rohu, Labeo rohita (Hamilton)"

_animals, 2023, doi:10.3390/ani13162631_

Round 1
Reviewer 1 Report
The study by Sukul and colleagues is an interesting work regarding pro / pre – biotics in fish. It is a well-designed and focused study and the authors have done a great job interpreting their results. Therefore, this article deserves publication in Animals MDPI. I have only minor comments / suggestions for the authors. These can be found in the attached pdf file.

Author Response
Itemized response to the Reviewer # 1
- "were acclimated" fits probably better here.
Ans. Revised as suggested.
- Please include in the footnote of the table abbreviations for MOC, CMC and NFE.
Ans. Abbreviations of MOC, CMC and NFE have been expanded in the Table 1.
- From the "Materials and Methods" I expected more biological replications. However, I understand the complexity of the experimental protocol, and the fact that n = 3 can be employed.
Ans. Thanks for the understanding.
- Please include in all figure legends what lower case letters stand for.
Ans. Lower case letters have been explained in the legend of Figures 1 and 2.
- The authors have done a great and complicated job here, gathering all these results and concluding to the overall figure of this species fitness. I would suggest the authors to employ a concluding figure (something like a graphical abstract) here in the "Conclusions" section in order for an accurate "take home message" for the readers.
Ans. Thanks for the encouraging comments. The graphical abstract was prepared and submitted during the first submission.
Reviewer 2 Report
The article is well written with all parts adequately described. The subject is very interesting and brings up a current trend in the use of food additives in order to improve the immunity of fish facing the challenges that occur in fish farms. There are some points that were not clear, and I would like them to be better explained:
- Was the ration used in the experiments the same as in the acclimatization period? There is a certain doubt in the text.
- In hematological analyses, how much blood was obtained from each fish? Due to the initial and final length and weight, it is presumed that they were small and that the amount of blood would not be sufficient for all analyses. Explain how they were performed, or how much blood was taken from each fish.
- In Table 1 put the meaning of the abbreviations: FCR,PER,ANPU.
- Tables 4, 5, 6 and 7 do not show the values of the analyzes on day 0, as shown in material and methods.
- In Table 4 the TEC (Total Erythrocyte Count) and TLC (Total Leucocyte Count), the best would be RBC (red blood cells) and WBC (white blood cells), which are universally used. In fact, the currently used VCM unit is fento liter (fL).
- The conclusions could be reduced because some statements are results and not conclusions.
Author Response
Itemized response to the Reviewer # 2
The article is well written with all parts adequately described. The subject is very interesting and brings up a current trend in the use of food additives in order to improve the immunity of fish facing the challenges that occur in fish farms. There are some points that were not clear, and I would like them to be better explained:
Was the ration used in the experiments the same as in the acclimatization period? There is a certain doubt in the text.
Ans. It has been mentioned in the “2.4 Experimental design” section that the fish were fed experimental diets twice daily (09:00 and 14:00 h) at 3% of their total biomass for 90 days. The feeding rate was adjusted every 15th day by batch-weighing.
In hematological analyses, how much blood was obtained from each fish? Due to the initial and final length and weight, it is presumed that they were small and that the amount of blood would not be sufficient for all analyses. Explain how they were performed, or how much blood was taken from each fish.
Ans. As indicated in the “2.8 Collection of blood samples” section, pooled samples of 6 fish were used for each replicate, and there were three replicates for each parameter. Further, as amount of blood was less from the small fishes, tail ablation method was followed to collect blood samples from the caudal vein.
In Table 3 put the meaning of the abbreviations: FCR, PER, ANPU.
Ans. Abbreviations are expanded at the footnote.
Tables 4, 5, 6 and 7 do not show the values of the analyzes on day 0, as shown in material and methods.
Ans. Initial values are included at the footnote.
In Table 4 the TEC (Total Erythrocyte Count) and TLC (Total Leucocyte Count), the best would be RBC (red blood cells) and WBC (white blood cells), which are universally used. In fact, the currently used VCM unit is fento liter (fL).
The conclusions could be reduced because some statements are results and not conclusions.
Ans. TEC (Total Erythrocyte Count) and TLC (Total Leucocyte Count) are replaced with RBC (red blood cells) and WBC (white blood cells), as suggested.
Reviewer 3 Report
Journal Animals (ISSN 2076-2615)
Manuscript ID animals-2437700
Type Article
Title Autochthonous Bacilli and Fructooligosaccharide as Functional Feed Additives Improve Growth, Feed Utilisation, Haemato-Immunological Parameters and Disease Resistance in Rohu, Labeo rohita (Hamilton)
Authors Tanaya Sukul , Zulhisyam Abdul Kari , Guillermo Téllez-Isaías , Koushik Ghosh *
This paper talks about the use of additives such as Fructooligosaccharide (FOS) and Bacillus spp. to fish feeding in order to improve growth performance, food conversion rate, immunological parameters and disease resistance. They used as a model the main carp specie produced in India Labeo rohita (Hamilton).
It is indeed a very interesting paper, with a lot of data from different tissues collected and very well performed.
I two major issues to discuss:
1. I have been working with fish and it shocked me the little error they had in their measurements, specially at Table 3 and Table 6. These tables hold data about growth performance and immune response variables (serum lysozyme, serum peroxidase or phagocytosis responses). Why just 3 fish if you had 30 fish in each tank plus 3 replicates per condition? How did you picked this 3 fish? Were the same 3 fish in all the tables? If you tagged the fish to recognize them please add it in Material and methods.
2. Symbiont effect is not a synergic effect. You must change it throughout the text. Symbiont condition is when two living organisms live together because one or both take advantage of that condition (mutualism, commensalism or parasitism). In this case, using bacillus could be a mutualism with the fish if it would work as a fish gut microbiota. However, you said symbiont as a synergic effect, meaning that pro- and prebiotic supplements work better when they are together than by separate.
I believe that is a good paper, with a lot of work behind, and if you correct these two things and few grammar mistakes, the paper can be easily improved and ready to publish.
In general, the paper is understandable, however there are small grammar mistakes like a lack of prepositions and wrong spelling. I strongly recommend to review the paper using any app that can show you the grammar mistakes quickly.
Author Response
Itemized response to the Reviewer # 3
This paper talks about the use of additives such as Fructooligosaccharide (FOS) and Bacillus spp. to fish feeding in order to improve growth performance, food conversion rate, immunological parameters and disease resistance. They used as a model the main carp specie produced in India Labeo rohita (Hamilton).
It is indeed a very interesting paper, with a lot of data from different tissues collected and very well performed.
I have two major issues to discuss:
- I have been working with fish and it shocked me the little error they had in their measurements, specially at Table 3 and Table 6. These tables hold data about growth performance and immune response variables (serum lysozyme, serum peroxidase or phagocytosis responses). Why just 3 fish if you had 30 fish in each tank plus 3 replicates per condition? How did you picked this 3 fish? Were the same 3 fish in all the tables? If you tagged the fish to recognize them please add it in Material and methods.
Ans. Please note that all surviving fish were collected from each tank and weighed to analyse the growth parameters. There were 3 replicates for each measurement/parameter (n = 3), however not analysed with only 3 fish. As indicated in the “2.8 Collection of blood samples” section, pooled samples of 6 fish were used for mucus, blood, head kidney and hepatic tissues for each replicate. For better comprehensibility, the things are clarified in the revised MS.
- Symbiont effectis not a synergic effect. You must change it throughout the text. Symbiont condition is when two living organisms live together because one or both take advantage of that condition (mutualism, commensalism or parasitism). In this case, using bacillus could be a mutualism with the fish if it would work as a fish gut microbiota. However, you said symbiont as a synergic effect, meaning that pro- and prebiotic supplements work better when they are together than by separate.
Ans. Regarding use of the terms "symbiotic" or "synbiotic", we fully agree with the comments of the Reviewer 3. However, we are surprised to see such a comment as we never used the term "symbiotic" for our work. Synbiotics refer to a combination of prebiotics and probiotics. In our study, we have used Fructooligosaccharide (FOS) as a prebiotic and B. licheniformis / B. methylotrophicus as the probiotics. Thus, the combination of FOS and B. licheniformis / B. methylotrophicus has been referred to as a “synbiotic" formulation in our MS.
I believe that is a good paper, with a lot of work behind, and if you correct these two things and few grammar mistakes, the paper can be easily improved and ready to publish.
Comments on the Quality of English Language
In general, the paper is understandable, however there are small grammar mistakes like a lack of prepositions and wrong spelling. I strongly recommend to review the paper using any app that can show you the grammar mistakes quickly.
Ans. Thanks for the encouraging suggestions. We have revised the MS for further clarity and comprehensibility.
Reviewer 4 Report
Dear Authors,
This study aimed to investigate the effects of the dietary administration of FOS and Bacillus spp. (B. licheniformis and B. methylotrophicus), both individually and in combination as synbiotics, on growth, haemato-immunological parameters and disease resistance capacity against pathogenic A. hydrophila in rohu, Labeo rohita, fingerlings.
The introduction provides a good, generalized background of the topic that quickly gives the reader appreciation of the scientific relevance and timeliness of the research theme.
- I think the findings of this study are sufficiently described in the context of the published literature. The conclusions are supported by appropriate evidence. However, too many references and among them too many old references (a little older than 10 years).
The manuscript presents an interesting topic. It also has a scientific value. However, some mistakes must be fixed before the publication.
I have some suggestions for Authors which are as follows:
- Table 1: the proximate composition shouldn’t be presented as % .
- The count of bacteria should presented as [log cfu/g].
- What probiotic properties did used strains have?
- Figure 4: unreadable. It needs to be corrected.
- Please thoroughly check the whole article and remove its grammatical mistakes.
- Recheck references according to the journal guidelines.
The article clearly meets the requirements of the scientific text. It is formally valid.
From my standpoint, this manuscript is appropriate for publication in Journal – Animals, after minor revision, given the above aspects.
Author Response
Itemized response to the Reviewer # 4
This study aimed to investigate the effects of the dietary administration of FOS and Bacillus spp. (B. licheniformis and B. methylotrophicus), both individually and in combination as synbiotics, on growth, haemato-immunological parameters and disease resistance capacity against pathogenic A. hydrophila in rohu, Labeo rohita, fingerlings.
The introduction provides a good, generalized background of the topic that quickly gives the reader appreciation of the scientific relevance and timeliness of the research theme.
I think the findings of this study are sufficiently described in the context of the published literature. The conclusions are supported by appropriate evidence. However, too many references and among them too many old references (a little older than 10 years).
Ans. Considering the complexity of the experimental protocol with numerous parameters, we had to refer some of the old but original methodology papers. In addition, some of the novel articles on the topic addressed in this MS were also referred to discuss the related issues. We hope that Hon’able Reviewer will be kind to retain the cited references for comprehensiveness of the article.
The manuscript presents an interesting topic. It also has a scientific value. However, some mistakes must be fixed before the publication.
I have some suggestions for Authors which are as follows:
- Table 1: the proximate composition shouldn’t be presented as % .
Ans. It is a convention that proximate compositions of the ingredients / diets are presented as % dry matter basis, if not specified otherwise (e.g., per kg). As other Hon’ble Reviewers didn’t make any comment on this issue, it has been retained.
- The count of bacteria should be presented as [log cfu/g].
Ans. Probiotic strains were added at 1×107 CFU/g diet (7 Log CFU/g). It has been mentioned in the text.
- What probiotic properties did used strains have?
Ans. It has been mentioned in the text (Introduction section) that the probiotic strains used in this study were bile-tolerant, diverse exo-enzymes producers and antagonistic against pathogenic aeromonads.
- Figure 4: unreadable. It needs to be corrected.
Ans. The referred figure has been supplied separately, may used during formatting and page setting.
- Please thoroughly check the whole article and remove its grammatical mistakes.
Ans. Thanks for your comment. Re-checked and revised accordingly.
- Recheck references according to the journal guidelines.
Ans. Thanks for your comment. Re-checked and revised accordingly.
The article clearly meets the requirements of the scientific text. It is formally valid.
From my standpoint, this manuscript is appropriate for publication in Journal – Animals, after minor revision, given the above aspects.
Ans. Thanks for the encouraging comments.